# MULTI-TASK LEARNING FOR ROUTING PROBLEM WITH ZERO-SHOT GENERALIZATION

## ABSTRACT

Vehicle routing problems (VRPs) are widely studied due to their significant practical importance. In the last decade, leveraging neural networks to solve VRPs in an end-to-end manner has gained substantial research attention. However, current works require building separate neural models for each routing problem, which hinders its practicality in solving diverse problems. In this study, we treat the VRPs as different combinations of a set of shared underlying attributes and propose to solve them simultaneously as multi-task learning. By training a unified model on multiple VRPs with varying attributes, we can effectively solve unseen problems in a zero-shot manner. Our experimental results on eleven VRPs show that our unified model performs comparably to single-task models trained specifically for each problem. More importantly, our model exhibits promising zero-shot generalization to new VRPs, reducing the average gap to 4.6% and 7.0% for sizes 50 and 100, respectively, compared to over 20% in the single-task approach.

## 1 INTRODUCTION

The vehicle routing problem (VRP) is a crucial topic in combinatorial optimization and operation research. It is widely studied in academia and has significant practical importance in real-world applications such as logistics, transportation, retail distribution, waste collection, and manufacturing (Toth & Vigo, 2014). The objective of the VRP is to manage a fleet of vehicles optimally, minimizing the total cost while satisfying the demands of customers. Real-world routing problems are typically subject to diverse attributes, which result in numerous VRP variants (Braekers et al., 2016; Vidal et al., 2020). Developing one algorithm for each VRP is very costly and impractical. Therefore, it is desirable to build a single unified solver for solving VRPs, which can significantly reduce the overall management cost and improve operational efficiency. A few attempts have been made to develop a unified VRP solver (Vidal et al., 2013; Rabbouch et al., 2021; Errami et al., 2023), but they demand much effort with domain knowledge from experts.

Neural combinatorial optimization (NCO) learns a heuristic based on neural networks for solving combinatorial optimization problems. This approach has received growing research attention due to its potential ability to generate high-quality solutions without much human effort (Bengio et al., 2021; Vinyals et al., 2015; Kool et al., 2018). However, most existing NCO approaches work in a single-task manner (Li et al., 2022; Bai et al., 2023). In other words, they need to train a neural network for each optimization problem. When the problem changes, e.g., the objective changes and/or new attributes are introduced, another neural network model needs to be trained from scratch, which inevitably leads to high computational costs. Some attempts to overcome this shortcoming include transfer learning and multiobjective learning (Feng et al., 2020; Li et al., 2021a; Zhang et al., 2022; Lin et al., 2022). However, the neural network models generated by these works can only be used to solve problems whose instances have been used for training. In other words, generalization across different problems has not been well addressed in the NCO community.

In this work, we take vehicle routing problems as a test bed and investigate whether a neural network model can solve a combinatorial optimization problem whose instances have not been included in the training dataset. We design a single neural network model to handle multiple VRPs which can be efficiently trained by reinforcement learning (RL) without labeled solutions. We show that the learned model can be used to solve VRP variants that have not been considered in the training in a zero-shot manner. Our contributions are summarized as follows:

- We propose a novel learning-based method to tackle cross-problem generalization in VRPs. It treats the VRP variants as different combinations of a set of shared underlying attributes, and solves various VRPs simultaneously in an end-to-end multi-task learning manner.
- We develop a unified attention model with a novel attribute composition block to handle multiple attributes for different VRPs. The proposed structure has a promising zero-shot generalization ability to handle any combination of the basic attributes. Many routing problems are solved for the first time using end-to-end NCO in this paper.
- We achieve competitive results with the existing single-task methods, which train one model for each problem. We also demonstrate promising zero-shot generalization capabilities on unseen VRPs. We reduce the average gap to 4.6% and 7.0% for sizes 50 and 100, respectively, from over 20% in the single-task approach.

## 2 RELATED WORK

**Neural combinatorial optimization (NCO)**  NCO (Bengio et al., 2021; Vinyals et al., 2015; Kool et al., 2018) intends to automatically learn a heuristic based on neural networks for solving the combinatorial optimization problem. Compared to the other approaches (e.g., exact methods and heuristics), it requires very little domain-specific knowledge and usually generates high-quality solutions significantly fast. As a result, it has gained much attention in the past decade (Bengio et al., 2021).

There are mainly two groups of works along this line: end-to-end methods (Vinyals et al., 2015; Bello et al., 2016; Nazari et al., 2018; Kwon et al., 2020; Joshi et al., 2022; Choo et al., 2022; Pan et al., 2023) and improvement-based methods (Chen & Tian, 2019; Hottung & Tierney, 2019; Chen & Tian, 2019; Kool et al., 2022). The former aims to construct a solution without any assistance from non-learning methods, while the latter incorporates additional algorithms to improve performance. In this paper, we focus on the end-to-end approach.

**NCO for vehicle routing problem (VRP)**  NCO has been successfully applied to many vehicle routing problems, including traveling salesman problem (TSP) (Bello et al., 2016), capacitated VRP (CVRP) (Nazari et al., 2018), VRP with time windows (VRPTW) (Zhao et al., 2020), open VRP (OVRP) (Tyasnurita et al., 2017), VRP with pickup and delivery (Li et al., 2021b), and heterogeneous VRP (Li et al., 2021a). A recent survey of the works on learning-based methods for different vehicle routing problems can be found in Li et al. (2022).

Despite extensive studies, the existing works have been conducted in a single-task manner, in which an individual neural model is trained for each problem. The time-consuming training process for every new problem hinders their practical application. It should be noted that various VRPs have common features, including shared objectives and underlying attributes. Nonetheless, these similarities and correlations have not been adequately studied in the context of NCO. Recently, Jiang et al. (2022); Bi et al. (2022); Geisler et al. (2022) explored the robust optimization over multiple distributions, several works (Fu et al., 2021; Pan et al., 2023; Manchanda et al., 2023; Cheng et al., 2023; Drakulic et al., 2023; Gao et al., 2023) studied generalization to large-scale problems and Zhou et al. (2023) considered generalization in terms of both problem size and distribution. The cross-problem generalization has not been studied in VRPs.

**Multi-task learning and zero-shot Learning**  Multi-task learning (MTL) tackles multiple related learning tasks in a single learning process. It has been widely studied in various research fields, including computer vision (Yuan et al., 2012), bioinformatics (He et al., 2016a), and natural language processing (Collobert & Weston, 2008). However, MTL has received limited attention on combinatorial optimization problems. Reed et al. (2022) and Ibarz et al. (2022) proposed a general agent capable of solving diverse tasks, including several combinatorial optimization problems. Wang & Yu (2023) presented a multi-task learning method for combinatorial optimization problems with separate encoders and decoders. Nevertheless, their approach falls short with respect to handling complicated VRPs, and they require revision or fine-tuning to solve new problems not previously encountered during training.

Zero-shot learning (ZSL) allows the recognition of previously unseen objects based on their shared semantic properties or attributes (Oh et al., 2017; Xian et al., 2018; Ruis et al., 2021). Our idea

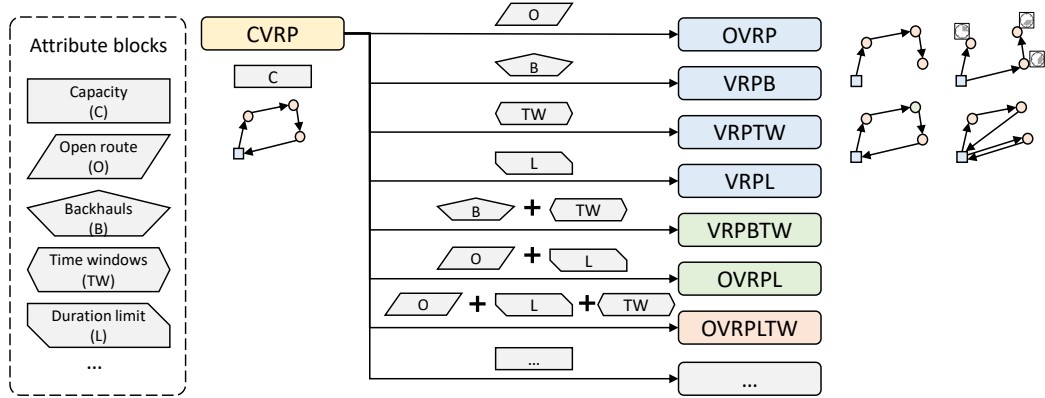

Figure 1: VRP variants as combinations of attribute blocks. The basic version is known as the CVRP. VRP variants can be regarded as extensions of CVRP, encompassing additional attributes. For instance, VRPTW extends CVRP by incorporating time windows, while OVRPTW adds open routes attribute alongside time windows.

of learning on multiple VRPs with several underlying attributes and generalizing to unseen VRPs is similar to compositional ZSL (Ruis et al., 2021), which composes novel problems out of known subparts or attributes.

## 3 PROBLEM STATEMENT AND MOTIVATION

We begin by introducing the formulation of the basic CVRP and then demonstrate that other VRP variants can be considered as extensions of the basic CVRP by incorporating additional attributes. We denote a CVRP on an undirected graph $G = (V, E)$. $V = \{v_0, \ldots, v_n\}$, where $v_0$ is the depot and $v_1, \ldots, v_n$ are the $n$ customers. $V_c = \{v_1, \ldots, v_n\}$ is the customer set. For the $i$-th customer, there is a demand $d_i$. $E = \{e_{ij}\}, i, j \in \{1, \ldots, n\}$ are the edges between every two nodes. For each edge $e_{ij}$, there is an associated cost (distance) $c_{ij}$. A fleet of homogeneous vehicles with a capacity of $C$ is sent out from the depot to visit the customers and return to the depot. Each customer must be visited once. The objective is to minimize the total traveling distance of all the used vehicles.

Figure 1 shows that various VRPs can be regarded as extensions of CVRP by considering one or more attributes. For example, VRPTW is extended from CVRP by adding time windows, and OVRPTW involves both time windows and open route attributes.

Except for the capacity constraints (C), we involve the following attributes in this paper:

- Time windows (TW): we denote the time windows $[e_i, l_i], i \in \{0, \ldots, n\}$ for the $i$-th node, where $e_i$ and $l_i$ are the early and late time windows. In addition, each node has a service time $s_i$. We consider hard time windows, i.e., the vehicle must visit the node $i$ in the time range from $e_i$ to $l_i$. If the vehicle arrives at node $i$ earlier than $e_i$, the vehicle has to wait until $e_i$.

- Open routes (O): open routes mean that the vehicle does not need to return to the depot after it services all the customers in its route.

- Backhauls (B): the classical CVRP assumes that all the vehicles load demands at the depot and unload at the customers. We call these customers, who require deliveries $d_i > 0$, linehaul customers. Correspondingly, backhaul customers are these customers that need pickup $d_i < 0$. We consider the VRP with mixed linehaul and backhaul customers, i.e., the order of linehaul and backhaul customers can be mixed up in each route.

- Duration limits (L): duration limits refer to the situation in which the total length of the routes can not exceed some pre-set thresholds. It is commonly used in real-world application scenarios to maintain a reasonable workload for different routes. In our setting, we use the same duration limit for all the routes.

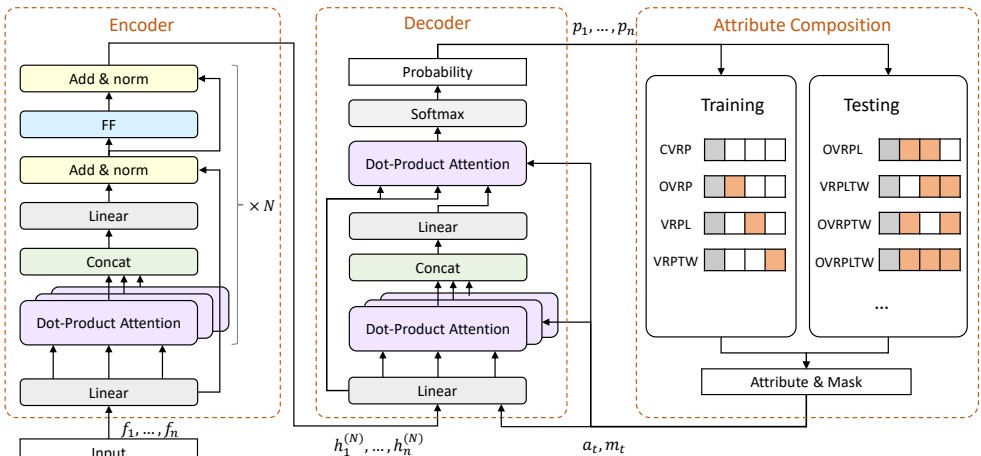

Figure 2: Unified model extended from attention model. The model is trained on multiple VRPs with diverse attributes. Then it can be used to solve numerous unseen VRPs as any combinations of the attributes involved in the training.

**Motivation** VRP variants can be extended from the basic CVRP by integrating one or more attributes. Even numerous VRPs involve only a few underlying attribute blocks, with the combinations of these blocks resulting in new VRPs. In previous works on NCO, the neural networks are trained independently for each routing problem without exploring potential similarities and correlations between different VRPs. In this paper, we aim to study the benefits of training a unified model on diverse VRPs as multi-task learning and demonstrate its capability of zero-shot generalization to new routing problems.

## 4 MULTI-TASK LEARNING FOR VEHICLE ROUTING

### 4.1 UNIFIED MODEL

We consider multiple VRPs as a set of related tasks and propose training a unified neural model through reinforcement learning to solve them simultaneously. Figure 2 illustrates the unified model used in this paper. It consists of three parts: encoder, decoder, and attribute composition. We adopt the typical encoder-decoder framework of attention model (Kool et al., 2018). The encoder learns the node embeddings, and the decoder generates solutions sequentially. Different from the existing works (Kool et al., 2018; Kwon et al., 2020; Zhu et al., 2023), we enable its ability to handle various VRPs by adding an additional attribute composition block. The idea is that the diverse VRPs actually consist of several common underlying attributes. By learning from these attributes, we can solve an exponential number of new VRPs as any combination of them.

**Encoder** The encoder consists of $N$ stacked multi-head attention (MHA) blocks (Kool et al., 2018). The input of the encoder is the node features $f_i, i = 1, \ldots, n$. In this paper, the input features for the $i$-th node are denoted as $f_i = \{x_i, d_i, e_i, l_i\}$, where $x_i$ are the coordinates, $d_i$ is the demand, and $e_i$ and $l_i$ are the early and late time windows, respectively. The input features are embedded through a linear projection to generate the initial feature embedding $h_i$. In each MHA layer, skip connections (He et al., 2016b) and instance normalization (IN) are used.

$$\hat{h}_i^{(l)} = IN^l \left( h_i^{(l-1)} + MHA^{(l)} \left( h_1^{(l-1)}, \ldots, h_n^{(l-1)} \right) \right)$$
$$h_i^{(l)} = IN^l \left( \hat{h}_i + FF^l \left( \hat{h}_i \right) \right) \tag{1}$$

where $l$ and $l-1$ represent the current and last MHA layers, respectively. The FF contains a hidden sublayer with ReLU activations. The above encoding process generates the final node embeddings

$h_i^{(N)}$. This encoding is performed only once, and the static node embeddings are reused for every decoding step.

**Decoder**  The decoder constructs a solution sequentially. The input of the decoder includes three parts: the node embedding $h_1^{(N)}, \ldots, h_n^{(N)}$, the embedding of currently visited node $h_t^{(N)}$, and the attribute embedding $a_t$. All the node embeddings are produced by the decoder. $a_t$ is the embedding of the current state of attributes. We provide an attribute vector to include various attributes involved in the multiple VRPs. At the $t$-th step, the attribute vector is $a_t = \{c_t, t_t, l_t, o_t\}$, where $c_t$ is the remaining capacity of the current vehicle, $t_t$ is the current time, $l_t$ is the current duration of the route, and $o_t$ indicates whether the route is open or not. Except for $o_t$, the others will be updated in each step. Backhauls are not embedded because they are implicitly considered in the node demands.

The decoder consists of one MHA layer and one single-head attention (SHA) layer with clipping. The MHA is slightly different from that used in the encoder. Skip connection, instance normalization, and FF sublayer are not used.

$$
\begin{aligned}
\hat{h}_c &= MHA_c\left(h_1^{(N)}, \ldots, h_n^{(N)}, h_t^{(N)}, a_t\right) \\
u_1 \ldots, u_n &= SHA_c\left(h_1^{(N)}, \ldots, h_n^{(N)}, \hat{h}_c\right)
\end{aligned}
\tag{2}
$$

The output embedding of MHA $\hat{h}_c$ is used as the input of the SHA, and the SHA outputs the probabilities of choosing the next node using a softmax $p_i = \frac{e^{u_i}}{\sum_j e^{u_j}}$. We omit the step indicator $t$ for readability. The detailed structure of the MHA and SHA can be found in Kool et al. (2018).

In each step, we need to mask some nodes from being selected. We update a masking vector $m_t$. The associated positions of unwanted nodes in the vector will be set to -inf, which will be used in the attentions before softmax. Except for masking those nodes that have already been selected in the previous steps, the infeasibility caused by various attributes should also be considered. For example, these nodes that violate time window constraints should not be selected.

**Attribute composition**  The input of attribute composition is the input node features $f_i, i = 1, \ldots, n$, the list of visited nodes $V_t$ at the current step $t$, and the problem attributes $A$. The output is the attribute vector $a_t$ and mask vector $m_t$.

The problem attributes $A$ are given explicitly with the input problem. $A$ are used to activate the corresponding attribute updating procedure in the attribute composition block. In this paper, we have four procedures for the four attributes. Each procedure $j$ updates the corresponding attribute in the attribute vector $a_t^j$ and calculates an infeasible node list that must not be visited in the next step $m_t^j$. The output attribute vector will include all the updated activated attributes and pad the inactivated attributes to be a default value. The output mask vector $m_t$ is the union of all activated attribute masks $m_t = V_t \cup \bigcup_{j \in A} m_t^j$. See Appendix A for the details of the attribute procedures.

For example, if only the capacity attribute is involved in the problem (i.e., CVRP), the indicator only activates the capacity updating procedure. In each step, the remaining capacity of the current vehicle is calculated. The attribute vector $a_t = \{c_t, t_t, l_t, o_t\}$ only updates $c_t$, and pads the rest attributes to zero. The infeasible nodes that exceed vehicle capacity when added into the route are updated to update masking $m_t = V_t \cup m_t^c$.

Most of the investigated VRPs involve subsets of attributes in our unified model. We are learning the shared underlying attributes of diverse VRPs. In this way, we can train on a few VRPs and solve a much larger group of VRPs as arbitrary combinations of the underlying attributes. This characteristic enables the zero-shot generalization ability of our model.

### 4.2 MULTI-TASK REINFORCEMENT LEARNING

We use the REINFORCE algorithm with a shared baseline following Kwon et al. (2020). We use greedy inference, i.e., a deterministic trajectory is constructed iteratively based on the policy. In each iteration, the next node is selected as the node with the maximum probability predicted by the decoder. $n$ trajectories are constructed from $n$ different starting points. Long-term

rewards $R(\tau_1), \ldots, R(\tau_n)$ (negative of the total distances) are calculated after the entire trajectories $\tau_1, \ldots, \tau_n$ are constructed. For the model with parameters $\theta$, the following gradient ascent is used:

$$\nabla_\theta J(\theta) \approx \frac{1}{nB} \sum_{i=1}^{B} \sum_{j=1}^{n} \left( R\left(\boldsymbol{\tau}_j^i \mid s_k^i\right) - b^i(s_k^i) \right) \nabla_\theta \log p_\theta \left(\boldsymbol{\tau}_j^i \mid s_k^i\right) \tag{3}$$

where $s_k$ represents the instances are generated from $k$-th task (VRP). $p_\theta(\boldsymbol{\tau}_j^i)$ is the aggregation of the probability of selection in each step of the decoder. $b^i(s_k) = \frac{1}{n} \sum_{j=1}^{n}(R\left(\boldsymbol{\tau}_j^i\right))$ is the shared baseline. $B$ is the batch size.

For multi-task learning, many optimizers have been designed to improve robustness and convergence. Instead of using the sophisticated multi-task optimizers (Chen et al., 2018; Kendall et al., 2018; Zhang & Yang, 2021), we simply trained the multiple tasks with equal weight.

## 5 EXPERIMENTS

We conduct experiments on eleven vehicle routing problems, namely CVRP, VRPTW, OVRP, VRPB, VRPL, VRPBTW, VRPBL, OVRPL, OVRPLTW, OVRPBTW, and OVRPBLTW. We train a unified model on the former five VRPs simultaneously and use the model to solve the rest problems in a zero-shot manner. Many of these questions are being addressed by the neural method for the first time. Note that one can easily extend the model to consider other attributes. We chose these attributes because they are among the most frequently used ones (Braekers et al., 2016).

**Instance generation** We use the same problem setup as that used in Kool et al. (2018) to generate the basic CVRP. For VRPTW, we use the method introduced in Zhao et al. (2020) to generate time windows and service times. For the rest problems, there is no existing work that solves exactly the same settings. We make the following settings: For VRPB, we first generate a CVRP and then randomly select 20% of customers as backhaul customers, whose demands are set to be the negative values of the original demands. For OVRP, we only need to set the open route indicator as active. For VRPL, the same maximum duration limit of 3 is used for each route.

**Model setting** The number of MHA for the encoder is 6, and the number of heads is 8. The hidden layer size is 512, and the embedding size is 128.

**Training** In training, we randomly select one type of VRP in each batch and generate instances of the selected VRP. We use 10,000 training instances for each epoch with a batch size of 64, and the number of epochs for training is 10,000. Adam optimizer is used. The initial learning rate is 1e-4 with a weight decay of 1e-6. We implement the unified model using PyTorch, and the experiments are running on a single RTX 2080Ti GPU. The training on the vehicle routing problems of size 100 costs about ten days.

Tabel 1 presents the required number of models and the total parameter sizes for single-task learning and our multi-task learning for five tasks. The required number of models (and the total parameters) could be much higher for single-task learning if we train a different model for each possible attribute combination, which is not affordable for real-world applications.

Table 1: Training costs between POMO with single-task learning and our model with multi-task learning on five VRPs

| | #Models | #Total Parameters | Instances per epoch | #Total Training Epochs | Training time cost (day) |
|---|---|---|---|---|---|
| POMO | 5 | 6.56M | 10,000 | 50,000 | 49 |
| Ours | 1 | 1.35M | 10,000 | 10,000 | 10.5 |

### 5.1 PERFORMANCE ON TRAINING VRPs

Table 2 lists the experimental results on the five VRPs used in the training. For CVRP, we compare our results to the original version of AM (Kool et al., 2018) and three extensions MDAM (Xin

et al., 2021), POMO (Kwon et al., 2020), and GCAM (Zhu et al., 2023). For VRPTW, a deep reinforcement learning method (DRL) (Zhao et al., 2020) and POMO are compared. The results of LHK3 (Helsgaun, 2017) and Gurobi are used as the baseline for CVRP and VRPTW, respectively. For the rest three problems, there are no existing end-to-end NCO methods for comparison. We implemented POMO on these problems and used the state-of-the-art hybrid genetic search (HGS) (Vidal et al., 2013) method as the baseline.

Table 2: Experimental results on five training VRPs. (The compared neural solvers require training one model for each VRP)

| Problem | Method | N=50 | | | N=100 | | |
| | | Dis. | Gap | Time | Dis. | Gap | Time |
|---------|--------|------|-----|------|------|-----|------|
| CVRP | HGS | 10.38 | - | 7h | 15.54 | - | 14h |
| | LKH3 | 10.38 | 0.00% | 7h | 15.61 | 0.46% | 14h |
| | AM (Samp1280) | 10.59 | 2.02% | 7m | 16.16 | 4.00% | 30m |
| | MDAM (BS50) | 10.48 | 0.96% | 7.5m | 15.99 | 2.90% | 26m |
| | GCAM (Samp1280) | 10.64 | 2.50% | - | 16.29 | 4.83% | - |
| | POMO | 10.53 | 1.41% | 3s | 15.87 | 2.13% | 10s |
| | POMO (Aug8) | 10.44 | 0.58% | 15s | 15.75 | 1.36% | 1.1m |
| | SGBS | 10.39 | 0.12% | 2.0m | 15.63 | 0.62% | 11.8m |
| | Ours | 10.56 | 1.73% | 3s | 15.90 | 2.29% | 11s |
| | Ours (Aug8) | 10.47 | 0.85% | 20s | 15.80 | 1.71% | 1.2m |
| | Ours+SGBS | 10.40 | 0.18% | 2.3m | 15.66 | 0.81% | 12.6m |
| VRPTW | HGS | 16.30 | - | 7h | 26.14 | - | 14h |
| | LKH3 | 16.52 | 1.36% | 7h | 26.60 | 1.76% | 14h |
| | DRL (BS10) | 17.90 | 9.82% | 1m | 29.50 | 12.85% | 2m |
| | DRL (BS10) +LNS | 16.94 | 3.93% | 11m | 27.44 | 4.97% | 65m |
| | POMO | 16.78 | 2.97% | 3s | 27.13 | 3.77% | 11s |
| | POMO (Aug8) | 16.66 | 2.22% | 19s | 26.91 | 2.93% | 1.2m |
| | SGBS | 16.55 | 1.52% | 2.9m | 26.55 | 1.58% | 15.1m |
| | Ours | 16.96 | 4.06% | 3s | 27.46 | 5.05% | 11s |
| | Ours (Aug8) | 16.80 | 3.09% | 20s | 27.13 | 3.81% | 1.2m |
| | Ours+SGBS | 16.58 | 1.71% | 3.2m | 26.63 | 1.89% | 17.9m |
| OVRP | HGS | 6.49 | - | 7h | 9.71 | - | 14h |
| | LKH3 | 6.52 | 0.46% | 7h | 9.75 | 0.41% | 14h |
| | POMO | 6.73 | 3.67% | 3s | 10.18 | 4.91% | 10s |
| | POMO (Aug8) | 6.63 | 2.14% | 16s | 10.07 | 3.76% | 1.1m |
| | SGBS | 6.56 | 1.12% | 2.1 m | 9.89 | 1.92% | 12.1m |
| | Ours | 6.81 | 4.90% | 3s | 10.34 | 6.56% | 11s |
| | Ours (Aug8) | 6.71 | 3.40% | 20s | 10.14 | 4.48% | 1.2m |
| | Ours+SGBS | 6.59 | 1.58% | 2.5m | 9.94 | 2.38% | 13.4m |
| VRPB | HGS | 7.69 | - | 7h | 11.13 | - | 14h |
| | LKH3 | 7.70 | 0.18% | 7h | 11.29 | 1.40% | 14h |
| | POMO | 7.92 | 3.06% | 3s | 11.57 | 3.88% | 10s |
| | POMO (Aug8) | 7.84 | 2.05% | 15s | 11.43 | 2.68% | 1.1m |
| | SGBS | 7.78 | 1.22% | 1.9m | 11.31 | 1.59% | 11m |
| | Ours | 8.17 | 6.36% | 3s | 11.72 | 5.23% | 11s |
| | Ours (Aug8) | 7.87 | 2.40% | 20s | 11.53 | 3.58% | 1.2m |
| | Ours+SGBS | 7.78 | 1.25% | 2.1m | 11.36 | 2.06% | 11.7m |
| VRPL | HGS | 10.37 | - | 7h | 15.54 | - | 14h |
| | LKH3 | 10.37 | 0.03% | 7h | 15.61 | 0.43% | 14h |
| | POMO | 10.55 | 1.78% | 3s | 15.84 | 1.96% | 10s |
| | POMO (Aug8) | 10.46 | 0.91% | 16s | 15.72 | 1.14% | 1.1m |
| | SGBS | 10.40 | 0.30% | 2.3m | 15.64 | 0.66% | 13.1m |
| | Ours | 10.56 | 1.88% | 3s | 15.96 | 2.72% | 11s |
| | Ours (Aug8) | 10.47 | 0.98% | 20s | 15.80 | 1.66% | 1.2m |
| | Ours+SGBS | 10.40 | 0.33% | 2.6m | 15.67 | 0.83% | 14.3m |
| Average | POMO | 10.50 | 2.58% | 3s | 16.12 | 3.33% | 10s |
| | POMO (Aug8) | 10.41 | 1.58% | 16s | 15.97 | 2.37% | 1.1m |
| | SGBS | 10.34 | 0.86% | 2.24m | 15.81 | 1.28% | 12.6m |
| | Ours | 10.61 | 3.78% | 3s | 16.27 | 4.37% | 11s |
| | Ours (Aug8) | 10.46 | 2.14% | 20s | 15.81 | 3.05% | 1.2m |
| | Ours+SGBS | 10.35 | 1.01% | 2.5m | 15.85 | 1.59% | 14.0m |

Table 3: Zero-shot generalization performance on five new VRPs.

| VRP | Method | n50 | | n100 | | VRP | Method | n50 | | n100 | |
|---|---|---|---|---|---|---|---|---|---|---|---|
| | | Dis. | Gap | Dis. | Gap | | | Dis. | Gap | Dis. | Gap |
| VRPBL | HGS | 7.70 | - | 11.15 | - | OVRPLTW | HGS | 10.69 | - | 17.35 | - |
| | NI | 11.69 | 51.86% | 17.38 | 55.92% | | NI | 15.74 | 47.20% | 26.16 | 50.78% |
| | FI | 11.61 | 50.81% | 16.37 | 46.88% | | FI | 15.22 | 42.41% | 25.81 | 48.79% |
| | POMO_CVRP | 8.21 | 6.61% | 12.41 | 11.37% | | POMO_CVRP | 15.23 | 42.46% | 26.75 | 54.18% |
| | POMO_VRPTW | 13.43 | 74.45% | 17.86 | 60.27% | | POMO_VRPTW | 11.51 | 7.70% | 19.41 | 11.88% |
| | Ours | **7.97** | **3.48%** | **11.65** | **4.50%** | | Ours | **11.50** | **7.59%** | **19.34** | **11.50%** |
| OVRPL | HGS | 6.49 | - | 9.71 | - | OVRPBTW | HGS | 10.67 | - | 17.31 | - |
| | NI | 13.85 | 113.55% | 13.61 | 40.17% | | NI | 15.76 | 47.77% | 26.24 | 51.60% |
| | FI | 14.34 | 121.11% | 13.46 | 38.62% | | FI | 15.22 | 42.69% | 25.79 | 48.99% |
| | POMO_CVRP | 7.75 | 19.47% | 11.78 | 21.35% | | POMO_CVRP | 15.25 | 42.98% | 26.78 | 54.69% |
| | POMO_VRPTW | 10.80 | 66.54% | 18.62 | 91.78% | | POMO_VRPTW | 11.52 | 7.94% | 19.48 | 12.51% |
| | Ours | **6.69** | **3.10%** | **10.15** | **4.57%** | | Ours | **11.49** | **7.72%** | **19.32** | **11.61%** |
| VRPBTW | HGS | 16.43 | - | 26.31 | - | Average | HGS | 10.39 | - | 16.36 | - |
| | NI | 18.92 | 15.15% | 36.84 | 40.05% | | NI | 15.19 | 46.16% | 24.05 | 46.95% |
| | FI | 18.33 | 11.56% | 36.59 | 39.10% | | FI | 14.95 | 43.78% | 23.60 | 44.25% |
| | POMO_CVRP | 22.98 | 39.85% | 38.99 | 48.20% | | POMO_CVRP | 13.88 | 33.56% | 23.34 | 42.64% |
| | POMO_VRPTW | **16.63** | **1.22%** | 27.18 | 3.33% | | POMO_VRPTW | 12.78 | 22.93% | 20.51 | 25.34% |
| | Ours | 16.70 | 1.66% | **27.11** | **3.05%** | | Ours | **10.87** | **4.59%** | **17.51** | **7.03%** |

We extended the original POMO on these problems and trained these models independently on each problem with single-task learning. We keep the settings of the original paper (Kwon et al., 2020). For other compared single-task learning models, we select the best results from the corresponding papers. If additional inference techniques are used, such as beam search (BS) and sampling (Samp), their size is indicated in the parentheses following the method. We adopt additional data augmentation (Aug) following POMO (Kwon et al., 2020). Additionally, the advanced inference strategy simulation guided beam search (SGBS) has been investigated and integrated into our framework for better zero-shot generation performance.

For each problem, the experiments are conducted on 5,000 instances. We compare the performance with respect to three criteria: the average distance (Dis.), the gap of the average distance to the baseline results, and the total running time on 5,000 instances. In general, our unified model is competitive with the existing single-task NCO methods. On CVRP and VRPTW, the results are better than the various existing end-to-end methods except for POMO. The gap between our model and optimal baselines is less than 5% across all five VRPs tested, and our running time is significantly less than baseline methods. The average gap of our unified model on the five training VRPs can be further reduced to around 1% when integrating with SGBS, with an acceptable increase in inference time. Despite slightly inferior results compared to POMO and SGBS, we note that the latter requires training individual neural networks for each problem and does not generalize well on unseen problems.

## 5.2 ZERO-SHOT GENERALIZATION TO UNSEEN VRPs

We use our unified model to solve unseen VRPs in a zero-shot manner. The experiments are carried out on five VRPs (i.e., VRPBTW, VRPBL, OVRPL, OVRPLTW, and OVRPBTW).

We compare the results of our unified model with single-task models, two commonly used constructive heuristics, and the SOTA heuristic HGS. The constructive heuristics are the nearest insertion method and farthest insertion method. We extended the source code of HGS so that it is able to solve various problems (Vidal et al., 2013). The two single-task models are POMO trained on CVRP and VRPTW, respectively. We added the masking procedure in our unified model so that they are applicable to different VRPs. Our unified model and the two single-task models are used in a zero-shot way without any fine-tuning on the new VRPs.

Table 3 shows the zero-shot performance. The results are evaluated on 5,000 instances for each problem. Our unified model outperforms other methods including two heuristic methods except for HGS, which is specifically developed for VRPs. The two single-task models are inferior to our multi-task unified model. The deficiency of single-task models is more obvious in problems with very different attributes. For example, the model only trained on CVRP performs worse on

these problems involving time windows attributes, while the model trained on VRPTW has poor performance on VRPBL. The average gap of our model over five VRPs is 4.6% and 7% on VRPs of sizes 50 and 100, respectively.

## 5.3 MORE DISCUSSIONS

We conduct additional experiments on all eleven VRPs with size 50 using our unified model to study 1) a comprehensive evaluation of our multi-task learning approach, 2) the contribution of attribute composition, and 3) fine-tuning. Figure 3 compares the gap between the baseline HGS and five different models on the eleven VRPs, with 5,000 instances for each problem. The detailed results can be found in the Appendix.

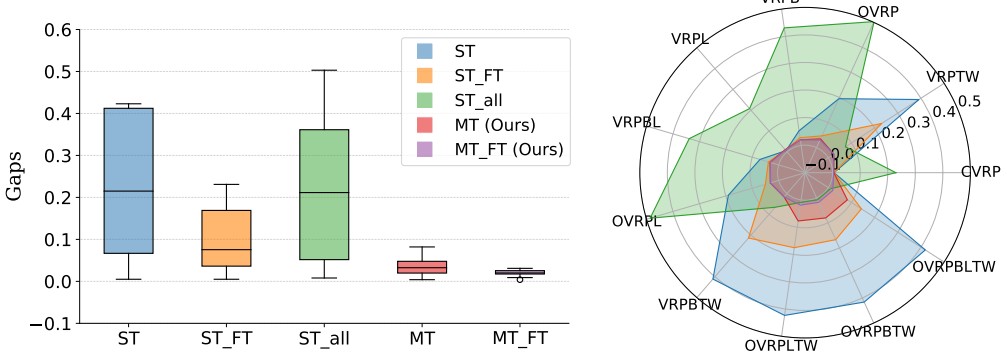

Figure 3: A comparison of gaps on eleven VRPs (Left: box plot, Right: radar plot). **ST** represents the unified model trained with single-task learning on CVRP, **ST_all** represents the unified model with single-task learning on OVRPBLTW, and **MT** represents our approach, i.e., the unified model with multi-task learning on five VRPs. **ST_FT** and **MT_FT** are the fine-tuning models.

**Overall performance** Our unified model with multi-task learning significantly outperforms single-task learning models in terms of overall performance. The average gap from the baseline is less than 4%, and this difference can be further reduced to 2% through fast fine-tuning. Furthermore, our method demonstrates strong generalization capabilities across different VRPs. In comparison, single-task learning models only perform well on the training problem. Additional results and comparisons can be found in the Appendix B. We also show that our cross-problem learning benefits cross-distribution generalization on out-of-distribution cases (Appendix D) and many benchmark test suites (Appendix E).

**Attribute composition** To demonstrate the advantages of multi-task learning with attribute composition, we train the unified model on OVRPBLTW with all attributes (ST_all). According to the results depicted in the radar plot, ST_all achieves promising performance on the problem it was trained on (OVRPBLTW), as well as on two VRP variants with similar attributes (OVRPBTW and OVRPLTW). However, its performance deteriorates significantly for the remaining VRPs. It suggests that direct training with all attributes fails to generalize effectively to other variants with only a subset of attributes. This comparison further confirms the advantage of learning different VRP variants through attribute composition.

**Fine-tuning** A fast fine-tuning of the unified model to each VRP can yield further performance improvement. Our unified model, when accompanied by fine-tuning, achieves the best results with a small gap across all VRPs. See Appendix C for the experimental settings of fine-tuning and more results.

We would like to highlight the possible impacts of our multi-task learning paradigm and zero-shot generalization ability for combinatorial optimization: 1) The design and implementation of SOTA heuristics on new problems require domain knowledge and much effort in development. Our unified model generates good solutions for unseen VRPs in a zero-shot manner. The development workload

for solving new VRPs is significantly reduced. 2) Any unseen VRP variants that are combinations of the learned attributes can be solved in an end-to-end way. In this way, we can train a unified model on a relatively small set of attributes and solve an exponential number of VRP variants. 3) In addition to routing problems, many other combinatorial optimization problems, such as scheduling problems and packing problems, can be regarded as combinations of common underlying attributes. Our approach is applicable to these problem sets.

## 6 CONCLUSION

This paper investigates multi-task learning for vehicle routing problems. The multiple VRPs are regarded as combinations of several shared underlying attributes. We propose to build a unified model to solve these related combinatorial optimization problems in an end-to-end way. The unified neural network is extended from the attention model with a unified encoder-decoder framework and attribute composition. To the best of our knowledge, solving multiple combinatorial optimization problems using a single neural network has seldom been studied, and many VRPs investigated in this paper are solved for the first time using the neural method. Experiments show that the unified model exhibits a promising zero-shot generalization ability on unseen VRPs that have not been used in training. Without any fine-tuning, it is significantly better than single-task models and outperforms two commonly used constructive heuristics. The average gap to baseline is reduced significantly from over 20% to 4.6% and 7% on sizes 50 and 100, respectively.

In the future, we would like to demonstrate multi-task learning on other combinatorial optimization problems, such as scheduling and packing. Moreover, other advanced multi-task learning techniques and approaches, e.g., prompt learning, can be investigated to further improve the zero-shot generalization performance of the unified model.

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

# A   MODEL DETAILS

## A.1   ATTENTION

We use the attention mechanism in Vaswani et al. (2017), which is a mapping (message passing (Kool et al., 2018)) of query $Q$, key $K$, and value $V$ vectors to an output. For each node $i$, the query $Q_i$, key $K_i$, and value $V_i$ are projections of the input embedding $h_i$:

$$Q_i = W^Q h_i, K_i = W^K h_i, V_i = W^V h_i. \tag{4}$$

where, parameters $W^Q$ and $W^K$ are of size ($d_k \times d_h$) and $W^V$ is of size ($d_v \times d_h$). We compute the compatibility $u_{ij}$ from the queries and keys as follows:

$$u_{ij} = \frac{Q_i^T K_j}{\sqrt{d_k}}. \tag{5}$$

We scale the compatibilities $u_{ij}$ using softmax to get attention weights $a_{ij} \in [0, 1]$:

$$a_{ij} = \frac{e^{u_{ij}}}{\sum_j e^{u_{ij}}}. \tag{6}$$

The output vector $h_i^o$ for node $i$ is the combination of the weights $a_{ij}$ and values $V_j$:

$$h_i^o = \sum_j a_{ij} V_j. \tag{7}$$

## A.2   MULTI-HEAD ATTENTION (MHA)

Multi-head attention enables the model to learn diverse information and usually benefits the results. MHA consists of $h$ heads and each head is an attention. It concatenates the results from all heads with a linear projection.

$$\begin{aligned} MHA(h_1, \ldots, h_n) &= Concat(head_1, \ldots, head_h)W^O \\ head_i &= Attention(h_1, \ldots, h_n) \end{aligned} \tag{8}$$

where $W^O$ has size ($hd_v \times d_k$). In our experiments, we use 8 heads with different parameters, and the embedding size is 128. For the attention model in each head, the parameter dimensions are $d_k = d_v = d_h/h = 16$.

## A.3   DECODER DETAILS

We use an MHA followed by a SHA in the decoder following Kool et al. (2018). The computing of queries, keys, and values for the MHA are as follows:

$$\begin{aligned} Q_c &= W^Q h_c, K_i = W^K h_i, V_i = W^V h_i, \\ h_c &= Concat(h_t, a_t), \end{aligned} \tag{9}$$

where $h_t$ is the embedding of the current visited node and $a_t$ is the attribute vector. $h_i$ is the output embedding from the encoder for node $i$.

In the SHA, we compute the compatibility using equation (5) and clip the results within [-10,10] with tanh. We also exclude the masked nodes by setting their compatibility values to -inf:

$$u_{cj} = \begin{cases} 10 \cdot tanh\left(\dfrac{q_c^T k_j}{\sqrt{d_k}}\right) & \text{if } j \notin m_t \\ -inf & \text{otherwise} \end{cases} \tag{10}$$

The output probability of selecting next node is computed as the softmax of the output compatibilities $p_i = \frac{e^{u_i}}{\sum_j e^{u_j}}$.

### A.4 ATTRIBUTE PROCEDURES

**Capacity** We track the remaining vehicle capacity $c_t$ at each step $t$, which is initially set to be the capacity of the vehicle $c_1 = 1$ (all the demands have been scaled by the capacity). After selecting a new node $v_t$, we update the remaining capacity as:

$$c_t = c_{t-1} - d_t \tag{11}$$

where $c_{t-1}$ represents the remaining capacity from the previous step, and $d_t$ represents the demand of the selected node in the current step $t$.

We mask these nodes that have already been visited or have demands that exceed the remaining vehicle capacity.

**Time windows** We keep track of the current time $t_t$ at each step $t$, initialized as $t_1 = 0$. After the selection of a new node, we update the current time as:

$$t_t = max(t_{t-1} + c_{(t-1),t}, e_t) + s_t \tag{12}$$

where $t_{t-1}$ represents the time from the previous step. $c_{(t-1),t}$ represents the distance between the last node and the current node (i.e., the traveling time cost between two nodes). $e_t$ and $s_t$ represent the early time window and the service time at node $v_t$, respectively.

We mask the visited nodes and the nodes whose time windows cannot be satisfied: 1) when we are unable to visit the node within the feasible time windows starting from the current node, or 2) when visiting the node would result in being too late to return back to the depot.

**Duration limit** We keep track of the route length $l_t$ at each step $t$, which is initialized to be zero $l_t = 0$. We update the current route length by adding the route length from the previous step $t - 1$ and the distance between the last node and the current node:

$$l_t = l_{t-1} + c_{(t-1),t} \tag{13}$$

We mask the nodes that exceed the duration limit when selected.

**Open route** We only need a fixed binary indicator for the open route attribute, with $o_t = 1$ representing the route is open and $o_t = 0$ otherwise. It has no contribution to the masking vector.

However, different from other attributes, the open route attribute results in a different total distance calculation. The distance between the last node and the depot is not included.

## B RESULTS ON ELEVEN VRPS

We present detailed results on eleven VRPs to demonstrate the advantages of our unified model with multi-task learning and attribute composition. We compared the following settings:

- Our unified model with multi-task learning.
- Our unified model with single-task learning on each training VRP.
- Our unified model with single-task learning on the VRP with all training attributes.

- POMO with single-task learning on each training VRP

We also examined the influence of the normalization method and included the results with fine-tuning. Table 4 lists all the results for VRPs of size 50. The top three results for each VRP are highlighted **in bold**. The abbreviations **ST** and **MT** represent single-task learning and our multi-task learning, respectively. **NN**, **BN**, **IN**, and **RN** denote no normalization, batch normalization, instance normalization, and re-zero normalization, respectively. **FT** represents fast fine-tuning. All the models were trained on a single RTX 2080Ti GPU. The testing time cost is the duration it takes to solve 5,000 instances for each VRP.

**MT vs. ST**   It is clear from this table that the single-task learning baselines perform similarly to the POMO counterparts. These results come without surprise since the POMO model serves as our base model. The performance of our multi-task learning model is slightly poorer than POMO and single-task learning (which is directly trained on each specific task), but our model has a much better (zero-shot) generalization performance on other seen/unseen tasks. Therefore, our method has a much better average performance across multiple VRP variants. In addition, we only need to build and train one single model to tackle all tasks with various attribute combinations, while single-task learning needs to build a different model for each task. The training budget has been significantly reduced.

**Attribute Composition**   As discussed in the main paper, we directly train the unified model on OVRPBLTW with all the attributes taken into account (ST_OVRPBLTW) to demonstrate the benefit of attribute composition. Line 11 in Table 4 lists the results of ST_OVRPBTW. According to the results, ST_OVRPBLTW can achieve promising performance on the problem it trained on (OVRP-BLTW) as well as two VRP variants with similar attributes (OVRPBTW and OVRPLTW). However, its performance becomes extremely poor for the rest VRPs (from simple CVRP to VRPBTW). This observation suggests that ST_OVRPBLTW is over-fitted to CVRPBLTW with all attributes and cannot generalize well to other variants with a subset of attributes. This comparison also confirms the effectiveness and usefulness of our proposed method for learning different VRP variants with attribute composition.

**Normalization**   Line 13 to 16 list the outcomes of the multi-task learning experiment without normalization, batch normalization, instance normalization, and re-zero normalization (BQ-NCO). Overall, we observed that normalization techniques had minimal impact on the results. Surprisingly, even without normalization, the results were already satisfactory. It is important to mention, however, that these tests were conducted solely on our unified model with VRPs of size 50. A comprehensive study is required in future work.

**Fine-tuning**   We conducted fast fine-tuning of ST_CVRP and MT on eleven VRPs. The fine-tuning on each problem costs less than two hours on size 50 and about 5 hours on size 100. The results indicate that the unified model with fine-tuning can yield further performance improvement. With fine-tuning, our unified model achieves the best results with a small gap across all VRPs. See Appendix C for the experimental settings of fine-tuning and more results.

Table 4: A summary of results on eleven VRPs with size 50. The top three results on each VRP are **in bold**. **ST** and **MT** represent single-task learning and the proposed multi-task learning, respectively. **NN**, **BN**, **IN**, and **RN** represent no normalization, batch normalization, instance normalization, and re-zero normalization, respectively. **FT** represents fast fine-tuning. All the models are trained on a single RTX 2080Ti GPU. The testing time cost is the time cost of solving 5,000 instances on each VRP.

| | | Method | Time Cost | | Gap | | | | | | | | | | | |
|---|---|---|---|---|---|---|---|---|---|---|---|---|---|---|---|---|
| | | | Training | Testing | CVRP | VRPTW | OVRP | VRPB | VRPL | VRPBL | OVRPL | VRPBTW | OVRPLTW | OVRPBTW | OVRPBLTW | Average |
| | | HGS | / | 7 h | 0 | 0 | 0 | 0 | 0 | 0 | 0 | 0 | 0 | 0 | 0 | 0 |
| | | LKH3 | / | 7 h | 0.05% | 1.29% | 0.94% | 0.40% | 0.96% | / | / | / | / | / | / | 0.73% |
| POMO | 1 | POMO_CVRP | 3.8 d | | **0.11%** | 38.93% | 19.57% | 5.38% | 0.83% | 6.65% | 19.65% | 39.54% | 42.66% | 42.48% | 43.22% | 23.55% |
| | 2 | POMO_VRPTW | 4.6 d | | 29.34% | **2.52%** | 66.48% | 58.91% | 28.06% | 53.55% | 66.66% | 1.60% | 7.70% | 7.91% | 8.00% | 30.06% |
| | 3 | POMO_OVRP | 4.3 d | 15 s | 9.28% | 45.48% | **2.11%** | 15.79% | 9.23% | 17.12% | 2.38% | 44.65% | 26.80% | 27.32% | 27.16% | 20.67% |
| | 4 | POMO_VRPB | 3.9 d | | 1.85% | 42.61% | 16.64% | **2.10%** | 1.90% | 3.05% | 16.75% | 43.90% | 42.60% | 41.55% | 41.77% | 23.16% |
| | 5 | POMO_VRPL | 3.9 d | | 0.49% | 38.73% | 19.70% | 5.48% | **0.57%** | 6.07% | 19.44% | 39.62% | 42.34% | 42.41% | 43.49% | 23.48% |
| Unified Model | 6 | ST_CVRP | 4.1 d | | **0.52%** | 39.11% | 19.52% | 5.37% | 0.59% | 7.07% | 19.02% | 41.10% | 42.34% | 41.64% | 41.88% | 23.47% |
| | 7 | ST_VRPTW | 4.8 d | | 39.15% | **2.16%** | 66.23% | 67.85% | 37.40% | 55.30% | 65.62% | **1.28%** | 7.83% | 7.99% | 7.91% | 32.61% |
| | 8 | ST_OVRP | 4.4 d | | 8.39% | 47.46% | **2.01%** | 16.24% | 8.49% | 17.03% | **2.12%** | 45.92% | 25.83% | 25.93% | 26.76% | 20.56% |
| | 9 | ST_VRPB | 4.1 d | 20 s | 1.22% | 41.50% | 16.94% | **1.80%** | 1.59% | 3.46% | 17.32% | 44.67% | 43.50% | 42.06% | 41.40% | 23.23% |
| | 10 | ST_VRPL | 4.3 d | | 0.52% | 38.42% | 19.56% | 8.44% | **0.44%** | 9.67% | 19.55% | 36.87% | 42.08% | 43.68% | 43.65% | 23.88% |
| | 11 | ST_OVRPBLTW | 5.4 d | | 22.95% | 7.40% | 50.29% | 43.11% | 20.77% | 33.84% | 48.63% | 6.62% | **0.90%** | **0.84%** | **0.86%** | 21.47% |
| | 12 | ST_CVRP (FT) | 4.1 d + 0.6 d | | / | 23.06% | 4.70% | 2.83% | 0.86% | 3.88% | 5.03% | 21.38% | 17.51% | 16.74% | 14.31% | 10.07% |
| | 13 | MT (NN) | 3.9 d | | 0.58% | 2.63% | 3.11% | 2.34% | 0.92% | 3.49% | 3.27% | 1.91% | 8.11% | 8.28% | 8.40% | 3.91% |
| | 14 | MT (BN) | 4.8 d | | 0.55% | 2.66% | 3.29% | 2.47% | 1.11% | 3.43% | 3.27% | 1.82% | 7.82% | **7.88%** | **7.82%** | **3.83%** |
| | 15 | MT (IN) | 4.8 d | 20 s | 0.42% | 2.42% | 3.50% | 2.05% | 1.07% | **3.28%** | **3.18%** | **1.64%** | **7.71%** | 8.95% | 8.05% | **3.75%** |
| | 16 | MT (RN) | 4.3 d | | 0.55% | 2.50% | 3.34% | 2.24% | 0.93% | **3.31%** | 3.35% | 2.19% | 7.82% | 8.08% | 8.13% | 3.86% |
| | 17 | MT (IN, FT) | 4.8 d + 0.6 d | | **0.37%** | **2.42%** | **3.04%** | **2.05%** | **0.86%** | **3.15%** | **3.06%** | **1.63%** | **1.90%** | **1.84%** | **1.96%** | **2.02%** |

## C   FAST FINE-TUNING ON UNSEEN VRPS

We conduct experiments to show the performance of our model under fast fine-tuning. The same five unseen VRPs are used: VRP with backhauls and time windows (VRPBTW), VRP with backhauls and duration limitation (VRPBL), open VRP with duration limitations (OVRPL), open VRP with duration limitations and time windows (OVRPLTW), and open VRP with backhauls and time windows (OVRPBTW).

Two different fine-tuning settings are tested: 1) only updating the decoder while keeping the encoder fixed, and 2) updating the entire model. Each epoch is trained using 10,000 instances with a batch size of 64, and 200 epochs are used. The learning rate and weight decay are set to 1e-5 and 1e-6, respectively. The experiments are carried out on the instances of size 100. The entire fine-tuning process with 200 epochs takes approximately five hours, while we note that the model typically converges within the first 50 epochs.

Figure 4 provides a comparison of different methods on the five VRPs in terms of distance. The detailed results are listed in Table 5, where the distance (Dis.), gap to the baseline HGS, and running time of methods are compared on 5,000 instances. The best results are shown in bold.

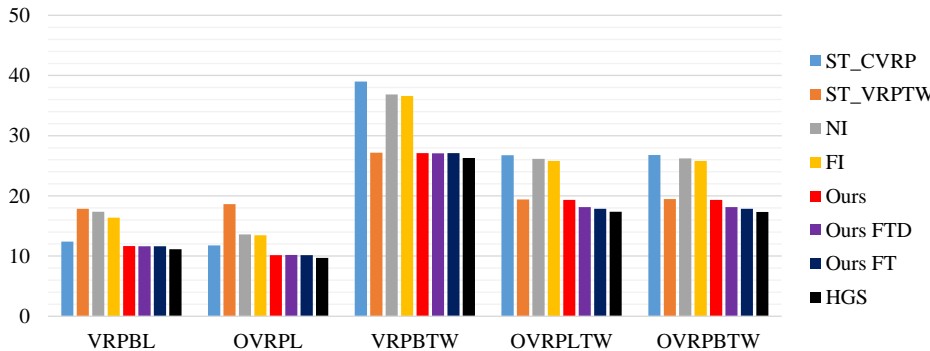

Figure 4: Comparison of the average results (total distances) of different methods on five new VRPs.

The abbreviations POMO_CVRP and POMO_VRPTW denote the single-task model (POMO) trained on CVRP and VRPTW instances, respectively. NI and FI represent the nearest and farthest insertion heuristics, and HGS is the SOTA algorithm for VRPs. FTD and FT indicate fine-tuning on the decoder and the entire network, respectively.

The results indicate that fine-tuning can further improve the performance of our model on new VRPs. Our pre-trained model with fast fine-tuning outperforms all other methods (except for the baseline). The average gap to the baseline HGS is only about 3.4% over the five VRPs. The advantages of fast fine-tuning become more apparent on OVRPLTW and OVRPBTW, which have more attributes and are therefore more complicated. In contrast, fine-tuning only provides minor improvements on VRPBL, OVRPL, and VRPBTW, where zero-shot generalization has already produced satisfactory results.

Figure 5 illustrates the convergence of testing distance vs. the number of fine-tuning epochs on OVRPLTW. We compare the performance of fine-tuning the pre-trained model with training a new model from scratch. The experimental settings of training from scratch are the same as that used for fine-tuning. The results demonstrate that fine-tuning based on our pre-trained unified model converges rapidly. The fine-tuning of the entire pre-trained model takes less than 50 epochs (about 1.5 hours) to converge, outperforming the results of training from scratch using 1,000 epochs. The advantage of fast adaptation highlights the significance of our pre-trained unified model.

The results demonstrate the promising generalization ability of our unified model on new problems. In addition to achieving overall acceptable results through direct zero-shot generalization, our proposed unified model can be fine-tuned on new problems with low computational cost, further improving the solution quality.

Table 5: Experimental Results on Five New VRPs.

| Problem | Method | Dis. | Gap | Time | Problem | Method | Dis. | Gap | Time |
|---|---|---|---|---|---|---|---|---|---|
| VRPBL | HGS | 11.15 | - | 14 h | OVRPLTW | HGS | 17.35 | - | 14 h |
| | NI | 17.38 | 55.92% | 8m | | NI | 26.16 | 50.78% | 8m |
| | FI | 16.37 | 46.88% | 8m | | FI | 25.81 | 48.79% | 8m |
| | POMO_CVRP | 12.41 | 11.37% | 1.1m | | POMO_CVRP | 26.75 | 54.18% | 1.1m |
| | POMO_VRPTW | 17.86 | 60.27% | 1.1m | | POMO_VRPTW | 19.41 | 11.88% | 1.1m |
| | Ours | 11.65 | 4.50% | 1.2m | | Ours | 19.34 | 11.50% | 1.2m |
| | Ours FTD | **11.62** | **4.27%** | 1.2m | | Ours FTD | 18.12 | 4.46% | 1.2m |
| | Ours FT | 11.64 | 4.47% | 1.2m | | Ours FT | **17.86** | **2.94%** | 1.2m |
| OVRPL | HGS | 9.71 | - | 14 h | OVRPBTW | HGS | 17.31 | - | 14 h |
| | NI | 13.61 | 40.17% | 8m | | NI | 26.24 | 51.60% | 8m |
| | FI | 13.46 | 38.62% | 8m | | FI | 25.79 | 48.99% | 8m |
| | POMO_CVRP | 11.78 | 21.35% | 1.1m | | POMO_CVRP | 26.78 | 54.69% | 1.1m |
| | POMO_VRPTW | 18.62 | 91.78% | 1.1m | | POMO_VRPTW | 19.48 | 12.51% | 1.1m |
| | Ours | 10.15 | 4.57% | 1.2m | | Ours | 19.32 | 11.61% | 1.2m |
| | Ours FTD | 10.18 | 4.87% | 1.2m | | Ours FTD | 18.12 | 4.70% | 1.2m |
| | Ours FT | **10.15** | **4.57%** | 1.2m | | Ours FT | **17.84** | **3.07%** | 1.2m |
| VRPBTW | HGS | 26.31 | - | 14 h | Average | HGS | 16.36 | - | 14 h |
| | NI | 36.84 | 40.05% | 8m | | NI | 24.05 | 46.95% | 8m |
| | FI | 36.59 | 39.10% | 8m | | FI | 23.60 | 44.25% | 8m |
| | POMO_CVRP | 38.99 | 48.20% | 1.1m | | POMO_CVRP | 23.34 | 42.64% | 1.1m |
| | POMO_VRPTW | 27.18 | 3.33% | 1.1m | | POMO_VRPTW | 20.51 | 25.34% | 1.1m |
| | Ours | 27.11 | 3.05% | 1.2m | | Ours | 17.51 | 7.03% | 1.2m |
| | Ours FTD | **27.08** | **2.94%** | 1.2m | | Ours FTD | 17.03 | 4.05% | 1.2m |
| | Ours FT | 27.11 | 3.05% | 1.2m | | Ours FT | **16.92** | **3.40%** | 1.2m |

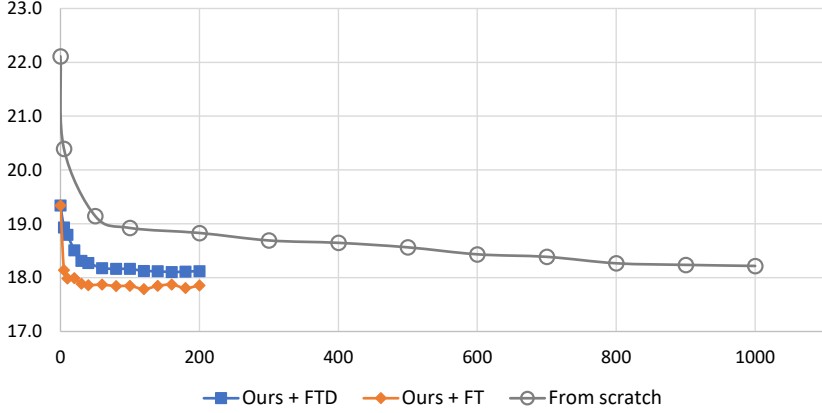

Figure 5: Testing distance vs. epoch number. FTD and FT indicate fine-tuning on the decoder and the entire network of our pre-train unified model, respectively. The curve with grey circles represents training a model from scratch. Each epoch includes 10,000 randomly generated OVRPLTW instances. The testing distance is the average result over 2,000 OVRPLTW instances.

# D EXPERIMENTS ON OUT-OF-DISTRIBUTION SCENARIOS

Existing works on neural combinatorial optimization typically assume that the training and testing instances come from the same distribution. Achieving generalization to out-of-distribution cases is often challenging for traditional single-task models. In this section, we investigate the generalization performance of our proposed multi-task model on three out-of-distribution scenarios of the training problems. All of our experiments are carried out on the problems of size 100.

## D.1 CVRP

**Basic settings:** The training instance generation for CVRP follows the distribution used in Kool et al. (2018). The coordinates of customers $x_i, y_i$ are randomly sampled in the unite region $[0, 1]$. The demands of customers $d_i$ are randomly selected from $\{1, \ldots, 9\}$ and then normalized with respect to the vehicle capacity $C$. The capacity is set to be $C = 40$ and $C = 50$ for the problem with a size of $n = 50$ and $n = 100$, respectively.

**Out-of-distribution settings:** We evaluate the performance of our model with POMO on CVRP instances with different vehicle capacities. Specifically, we consider seven vehicle capacities: $C = \{20, 30, 40, 50, 60, 70, 80\}$.

Table 6 and Figure 6 show the results on out-of-distribution CVRP instances averaged over 5,000 instances. The better results are in bold and the gap is calculated using POMO as the baseline. A small gap is preferred and a negative gap indicates that our model outperforms POMO. Except for the results on 50 and 70 (close to that used for generating training instances), our model beats POMO on all the out-of-distribution cases. The advantage of using our unified model is more obvious with the increasing vehicle capacity.

Table 6: A comparison of our model with single-task POMO on out-of-distribution CVRP instances

| Vehicle capacity | 30 | 50 | 70 | 90 | 110 | 130 | 150 | 200 |
|---|---|---|---|---|---|---|---|---|
| POMO_CVRP | 22.913 | **15.750** | **12.910** | 11.480 | 10.595 | 10.114 | 9.824 | 9.307 |
| Ours | **22.804** | 15.750 | 12.914 | **11.441** | **10.511** | **9.900** | **9.532** | **8.969** |
| Gap | -0.48% | 0.00% | 0.03% | -0.33% | -0.80% | -2.12% | -2.97% | -3.63% |

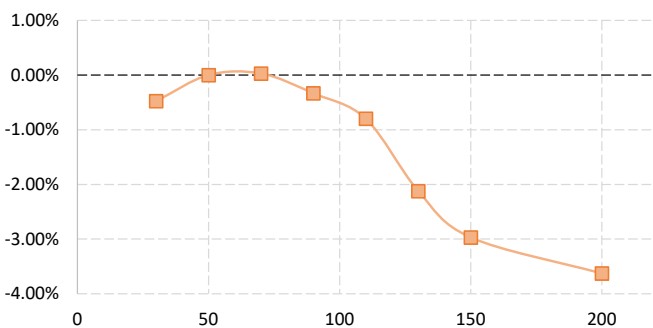

Figure 6: Performance gap between Ours and POMO with respect to different vehicle capacities on CVRP.

## D.2 VRPTW

**Basic settings:** For VRPTW, we adopt the same settings as those used in Zhao et al. (2020). Specifically, we generate the coordinates, demands, and vehicle capacities using the same procedure as for CVRP. The time windows attributes involve three additional features: 1) the service time $s_i$,

2) the early time $e_i$, and 3) the late time $l_i$. The service time $s_i$ and the length of the time window $\Delta_i$ are randomly sampled from a closed interval $[0.15, 0.2]$. The speed of the vehicle is fixed at $v = 1$ and the maximum time interval of the depot is set to $T = 4.6$. It means that all vehicles must return to the depot before the maximum time interval.

The early and late times for each customer can be calculated as follows:

$$e_i = \frac{h_i \times c_{0i}}{v}, h_i \in \left[1, \frac{T - s_i - \Delta_i}{c_{0i}} \times v - 1\right]$$
$$l_i = e_i + \Delta_i$$
(14)

where $c_{0i}$ denotes the distance between the depot and the $i$-th customer. This formulation makes sure that there is at least one feasible solution for each instance.

**Out-of-distribution settings:** For out-of-distribution testing, we only modify the closed time interval used for sampling service time and time window length. The intervals are as follows: $\{[0.05, 0.1], [0.15, 0.2], \ldots, [0.85, 0.9], [0.95, 1.0]\}$.

Table 7: A comparison of our model with single-task POMO on out-of-distribution VRPTW instances.

| Time Interval | [0.05,0.1] | [0.15,0.2] | [0.25,0.3] | [0.35,0.4] | [0.45,0.5] |
|---|---|---|---|---|---|
| POMO_VRPTW | **26.230** | **26.906** | **28.805** | 32.327 | 36.544 |
| Ours | 26.390 | 27.152 | 29.046 | **32.174** | **36.162** |
| Gap | 0.61% | 0.91% | 0.84% | -0.47% | -1.05% |

| Time Interval | [0.55,0.6] | [0.65,0.7] | [0.75,0.8] | [0.85,0.9] | [0.95,1.0] |
|---|---|---|---|---|---|
| POMO_VRPTW | 40.650 | 45.327 | 49.219 | 53.300 | 57.102 |
| Ours | **40.136** | **45.007** | **48.901** | **52.815** | **56.209** |
| Gap | -1.27% | -0.71% | -0.65% | -0.91% | -1.56% |

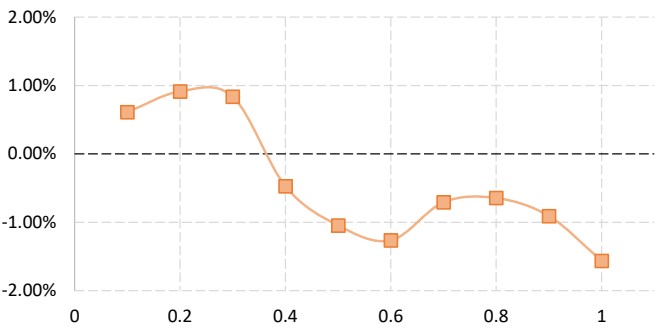

Figure 7: Performance gap between Ours and POMO with respect to different time intervals on VRPTW.

The results in Table 7 and Figure 7 again reveal that our model is more robust in out-of-distribution cases. We have negative gaps in the majority of cases except for these near training distribution $[0.15, 0.2]$.

### D.3  VRPL

**Basic settings:** Same as CVRP.

**Out-of-distribution settings:** In the training, the duration limit is fixed to $l = 3.0$. We modify the duration limit of each route. The settings used are $l = \{2.9, 3.0, 3.1, 3.2, 3.3, 3.4, 3.5\}$. To better

Table 8: A comparison of our model with single-task POMO on out-of-distribution VRPL instances.

| Duration | 2.9 | 3 | 3.1 | 3.2 | 3.3 | 3.4 | 3.5 |
|---|---|---|---|---|---|---|---|
| POMO_VRPL | 10.245 | 10.145 | 10.077 | 10.044 | 10.019 | 10.000 | 9.998 |
| Ours | **9.858** | **9.748** | **9.683** | **9.642** | **9.625** | **9.601** | **9.561** |
| Gap | -3.78% | -3.91% | -3.91% | -4.00% | -3.94% | -3.99% | -4.37% |

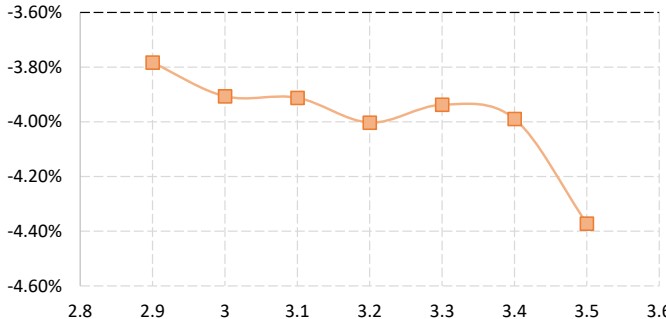

Figure 8: Performance gap between Ours and POMO with respect to different duration limitations on VRPL.

show the effect of the duration limit on the results, we raise the vehicle capacity from 50 to 150 for VRPL. In this way, the number of nodes in each route will increase and the influence of duration limit will be more significant.

Table 8 and Figure 8 show the results. Our model outperforms POMO on all the distributions (the reason might be that we have also modified the vehicle capacity). The advantage of our model is more obvious on large duration limits.

Our experiments reveal that our multi-task model displays a strong generalization performance across various distributions. Conversely, the single-task models perform best only on the test scenarios that share the same distributions as the training data. They are outperformed by the multi-task model when it comes to out-of-distribution scenarios.

We observed that a multi-task model's advantage becomes more pronounced in extreme cases. For instance, on CVRP with a vehicle capacity of 200, our model exhibits a substantial performance gap compared to POMO models. One possible explanation for this is that training a multi-task model with different VRP variants brings diversity. Although it does not specifically represent the utilization of various vehicle capacities, the multi-task model can extract more comprehensive patterns than single-task learning.

# E    EXPERIMENTS ON BENCHMARKS AND LARGE-SCALE INSTANCES

We validate our model on CVRPLib benchmarks as well as randomly generated large-scale instances. Specifically, we select six test suites with diverse attributes from CVRPLIB [1]. There are a total of 181 instances whose problem size ranges from 30 to 1,000. We normalize the coordinates of customers so that they are within the unit range of $[0, 1]$. We also normalize the demands with respect to the vehicle capacity.

To evaluate the performance, we compare our unified model with POMO, which is trained on CVRP. Both models are trained on the instances of size 100. We use the best-known solutions (BKS) provided by CVRPLIB as the baseline. The criteria for comparison include the average distance over all test suite instances, the average gap to BKS, and the standard deviation (SD) of the gap.

**A-P Test Sets**    Table 9 lists the information of six test suites as well as the experimental results. The results demonstrate that, although our model is not specifically trained for CVRP, it outperforms POMO on all test suites. Overall, the average gap between our unified model and the BKS is less than 10%, which is about half of POMO. In addition, our model has a lower standard deviation in the majority of cases, indicating it is more robust than POMO on diverse distributions.

**X Test Set**    More analysis of the results obtained from test suite X is provided. We divided the 100 instances in X into four groups based on their problem size. The results of our model and POMO on each group are summarized in Table 10. Our findings indicate that both models perform similarly well on small-scale instances, but as the scale of the problem increases, the performance of POMO deteriorates. In fact, in the largest group, the average gap of POMO is over 30%. Conversely, our model is more robust across problem sizes, with its gap increasing only from 6.2% to 13.8%.

Table 9: Experimental results on six test suites.

| Beckmark | No. | Size | BKS | POMO Dis. | POMO Gap | POMO SD | Ours Dis. | Ours Gap | Ours SD |
|---|---|---|---|---|---|---|---|---|---|
| A | 27 | 31-79 | 1041.9 | 1104.8 | 6.0% | 7.2% | 1066.8 | **2.4%** | **1.0%** |
| B | 23 | 30-77 | 963.7 | 1065.7 | 10.6% | 14.6% | 992.2 | **2.9%** | 4.6% |
| F | 3 | 44-134 | 707.7 | 770.6 | 8.9% | **1.1%** | 760.6 | **7.5%** | 3.3% |
| M | 5 | 100-199 | 1083.6 | 1145.2 | 5.7% | 3.0% | 1141.9 | **5.4%** | **2.6%** |
| P | 23 | 15-100 | 587.4 | 660.4 | 12.4% | 35.0% | 627.1 | **6.8%** | 6.1% |
| X | 100 | 100-1k | 63106.7 | 75154.8 | 19.1% | 15.2% | 69473.5 | **10.1%** | **4.7%** |
| Average | - | - | - | - | 14.7% | - | - | **7.4%** | - |

Table 10: Experimental results on test suite X with respect to different problem sizes.

| Size | No. | BKS | POMO Dis | POMO Gap | Ours Dis | Ours Gap |
|---|---|---|---|---|---|---|
| 100-300 | 43 | 33868.5 | 36299.1 | 7.2% | 35954.7 | **6.2%** |
| 300-500 | 15 | 63774.8 | 71524.1 | 12.2% | 68841.6 | **7.9%** |
| 500-700 | 15 | 88561.4 | 107927.9 | 21.9% | 98843.8 | **11.6%** |
| 700-1000 | 17 | 113619.5 | 149858.6 | 31.9% | 129270.9 | **13.8%** |

---

[1]http://vrp.atd-lab.inf.puc-rio.br/

**XML100 Instances** We have conducted additional experiments on XML100 test suites (Queiroga et al., 2022), which was proposed recently in AAAI 2022 for testing learning methods for vehicle routing. We compared the results to HGS, LKH3, and OR-Tools which can be found in Table 11. According to the results, our method has a good generalization performance on these problems. Our model has an average gap of 5.42%, which is inferior to LKH3 but much better than POMO and OR-Tools. Additionally, the running time of our model is less than 0.1s.

Table 11: A comparison of results on 30 XML100 instances.

| | BK Dis. | HGS (5 s) | | LKH3 (60 s) | | OR-Tools (60 s) | | POMO ($<$0.1s) | | Ours ($<$0.1s) | |
|---|---|---|---|---|---|---|---|---|---|---|---|
| | | Dis. | Gap | Dis. | Gap | Dis. | Gap | Dis. | Gap | Dis. | Gap |
| SET 1 | 13179 | 13182 | 0.02% | 13555 | 2.29% | 14780 | 16.01% | 16988 | 29.47% | 13946 | 6.23% |
| SET 2 | 10738 | 10740 | 0.01% | 10969 | 1.53% | 11923 | 12.28% | 14851 | 41.01% | 11318 | 5.60% |
| SET 3 | 18996 | 19028 | 0.17% | 19993 | 2.76% | 20791 | 12.31% | 20767 | 11.83% | 19867 | 4.44% |
| Average | 14304 | 14317 | 0.07% | 14839 | 2.19% | 15831 | 13.53% | 17535 | 27.44% | 15044 | 5.42% |

**CVRP200-500** We compare results on CVRP200-500

Table 12: Cross-size performance on randomly generated CVRP instances.

| | N200 | | N300 | | N400 | | N500 | |
|---|---|---|---|---|---|---|---|---|
| HGS | 21.82 | / | 25.76 | / | 28.14 | / | 32.01 | / |
| LKH3 | 22.34 | 2.39% | 26.45 | 2.72% | 29.05 | 3.23% | 32.07 | 0.17% |
| OR-tools | 24.14 | 10.63% | 28.82 | 11.89% | 31.77 | 12.89% | 35.82 | 11.90% |
| POMO | 23.11 | 5.90% | 29.21 | 13.42% | 34.75 | 23.48% | 43.12 | 34.68% |
| | 22.85 | 4.71% | 28.79 | 11.77% | 33.76 | 19.98% | 41.14 | 28.50% |
| Ours | 22.96 | 5.25% | 28.42 | 10.35% | 32.53 | 15.60% | 38.53 | 20.34% |
| | 22.73 | 4.18% | 28.07 | 8.97% | 32.02 | 13.78% | 37.45 | 16.99% |
| Ours SGBS | **22.34** | **2.40%** | **27.30** | **6.00%** | **30.92** | **9.87%** | **35.78** | **11.75%** |

**Visualization** Figure 9 illustrates the solutions generated by our model and POMO on two instances: X-n322-k28 and X-n561-k42. It is clear from the figure that the results of POMO display crossovers between different routes, while our model produces more structured solutions.

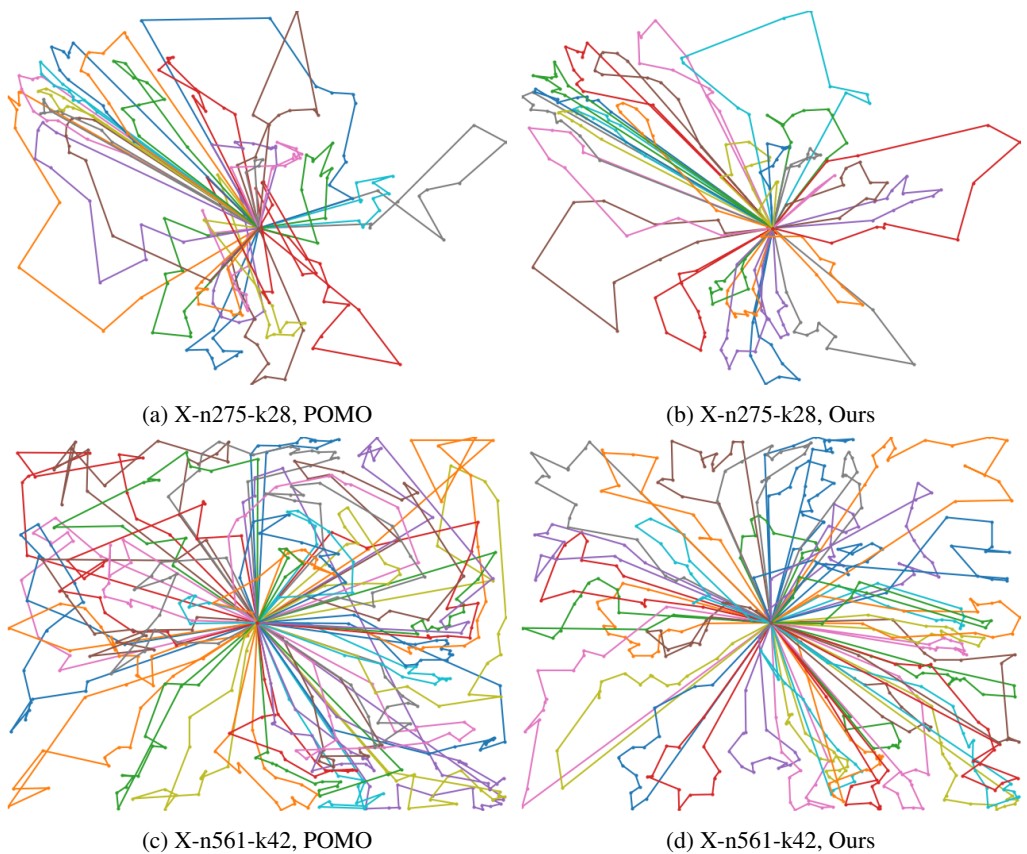

(a) X-n275-k28, POMO

(b) X-n275-k28, Ours

(c) X-n561-k42, POMO

(d) X-n561-k42, Ours

Figure 9: Illustration of the solutions generated by single-task POMO and our proposed multi-task model on X-n322-k28 and X-n561-k42 instances.

# F  ATTRIBUTE CORRELATION

**Correlation among Attributes:** Similar to Standley et al. (2020), we separately train our model on each pair of VRPs to investigate the pairwise correlation among tasks. As a result, we have a total $C(5, 2) = 10$ pairwise combinations with respect to the 5 basic VRPs (i.e., CVRP, VRPTW, OVRP, VRPB, and VRPL).

We fine-tune the 10 models from our pre-trained unified model, instead of training them from scratch. Each epoch is trained using 10,000 instances with a batch size of 64, and 200 epochs are used (the model typically converges within the first several epochs). The learning rate and weight decay are set to 1e-4 and 1e-6, respectively. The experiments are carried out on the instances of size 50.

The resulting correlation matrix is shown in Figure 10, which measures the performance gap between each task pair and our unified model. The positive values represent a positive correlation between two VRPs, while negative values denote a conflict.

The results reveal that the CVRP, being a fundamental routing problem, has a positive correlation with all the other problems except for OVRP. The OVRP is found to exhibit the most severe conflicts among the attributes followed closely by the VRPTW. It aligns with our intuition that the open route conflicts with the other VRPs that force every route back to the depot. Similarly, time windows pose additional constraints on the choosing of the next node, which may significantly influence generalization performance, as we will discuss in the next part.

**Distribution of Different VRPs:** We have also conducted a comparison of distributions of the hidden layers of different VRPs on the two-dimensional reduction space. Fig 11 shows a comparison of distributions of different VRPs on two-dimensional reduction space of the decoder hidden layer (1000 samples for each VRP). There is a clear distinct distribution of VRPTW compared with others as the time window attribute poses a strong constraint over the route. In addition to VRPTW, OVRP also shows a different pattern than others followed by VRPB, this is consistent with the correlation matrix that OVRP does not force routes back to the depot. In contrast, CVRP and VRPL follow a very close distribution and overlap with each other in the majority of areas, which is also reflected in the close final distances. This can be attributed to the fact that our route length limit is set to 3 on the normalized coordinates, which is easily satisfied due to the tighter capacity constraint. The distribution analysis of the hidden layer aligns with the findings from our correlation matrix discussions.

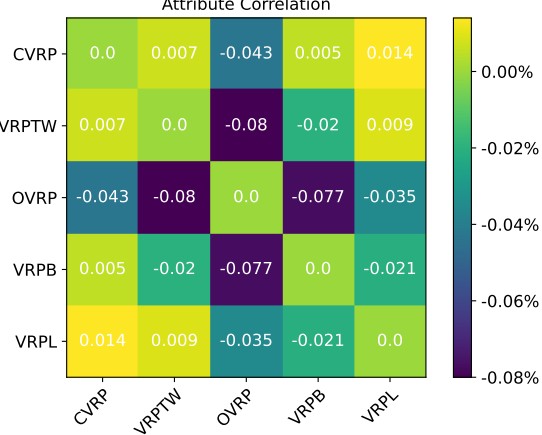

Figure 10: **Correlation among Attributes:** Positive values indicate a positive correlation between two attributes, and negative values represent conflict. It can be observed that the CVRP, being a fundamental routing problem, has a positive correlation with all the other problems except for OVRP. The OVRP exhibits the most severe conflicts, while the VRPTW follows closely behind.

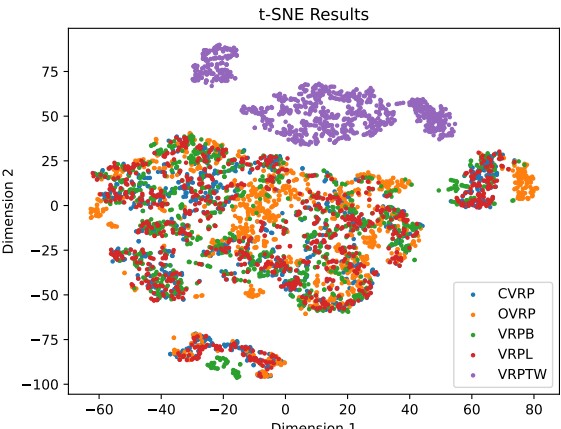

Figure 11: A comparison of distributions of different VRPs on two-dimensional reduction space of the decoder hidden layer.

**Influence of the Degree of Constraint:** In the second part of the experiments, we examine how the degree of constraint affects the correlation and zero-shot generalization performance. We illustrate this using an example of VRPTW (Vehicle Routing Problem with Time Windows). Specifically, we trained a single-task model on VRPTW (ST_VRPTW) with various TW lengths $\Delta_i = 0.15$, 1 ,2 ,4 (where a smaller $\Delta_i$ represents a tighter TW constraint). We then evaluated the models on OVRPLTW and OVRPBTW. Our results reveal that as the TW length increases (i.e., the time window constraint is relaxed), the performance gap becomes larger compared to our multi-task learning approach. This can be attributed to the fact that the time window constraint, which is tightly enforced in our paper, is a dominant factor compared to other attributes that explain the strong generalization of ST_VRPTW on tasks with time window constraints.

Table 13: A comparison of the zero-shot generalization performance of ST_VRPTW with different TWs and our MT

|  | ST_VRPTW | | | | MT |
|---|---|---|---|---|---|
|  | 0.15 | 1 | 2 | 4 |  |
| OVRPLTW | 7.83% | 10.71% | 14.85% | 17.07% | 7.71% |
| OVRPBTW | 7.99% | 11.55% | 15.46% | 17.46% | 7.95% |

