# OpenReview forum: "Multi-Task Learning for Routing Problem with Zero-Shot Generalization"
_ICLR.cc/2024/Conference — Submitted to ICLR 2024_

### Official Review · Reviewer_kE7F · 2023-10-16

**Soundness:** 3 good
**Presentation:** 3 good
**Contribution:** 2 fair
**Rating:** 6
**Confidence:** 4

**Summary:**

This paper presents a unified neural model based on multi-task learning for solving various VRPs. The underlying structure includes encoder, decoder, and attribute composition block. The experiments are conducted on 11 VRP variants, and showed effectiveness against the selected baseline methods.

**Strengths:**

1- The writing is good, and the paper is easy to follow;
2- This paper is well-motivated;
3- It is evaluated on eleven VRP variants.

**Weaknesses:**

1- While this work is well motivated, there is no novelty from the methodology perspective. It seems like simple adaption of multitask learning and AM. Although the authors claim the attribute composition is novel, it looks quite naive and trivial.
2- The baseline POMO is classic, but not the SOTA. Quite a number of subsequent works surpass it, such as Efficient Active Search and 'Simulation-guided Beam Search for Neural Combinatorial Optimization'. In this sense, the comparison is not that convincing.
3. Although the authors reviewed and criticized 'Efficient Training of Multi-task Combinarotial Neural Solver with Multi-armed Bandits', it is the most relevant baseline to this paper. It is good to compare with it in some adapted or tailored experimental settings.
4. In table 2, where the proposed method is inferior to other neural baselines should be analyzed and explained.
5. In table 3, the generic traditional methods like LKH-3 and OR-tools should be included.
Overall, the methodological novelty and the experimental results are not significant.

**Questions:**

Please see the above Weaknesses.

---

> ### Author Response · Authors · 2023-11-20
> **Response to Review kE7F part 1/3**
>
> Thank you very much for your time and effort in reviewing our work and providing valuable suggestions. We address your concerns as follows.
>
> > **W1. While this work is well-motivated, there is no novelty from the methodology perspective. It seems like simple adaption of multitask learning and AM. Although the authors claim the attribute composition is novel, it looks quite naive and trivial.**
>
>
> In this work, rather than the simple multi-task learning, we investigate the **attribute composition for NCO**, which is a significant but less explored nature of many combinatorial optimization problems such as VRPs [1]. By only learning from a small number of problems with five fundamental attributes (capacity, time windows, open routes, backhauls, and duration limits), our model can easily generalize to tackle a larger number of problems with various combinations of these attributes in a zero-shot manner. This zero-shot generalization property is significantly different from a simple MTL problem and has not been investigated for neural combinatorial optimization. Our proposed attribute composition surpasses simply training all attributes as well as single-task learning in zero-shot generalization.
>
> In addition, the implementation of both our unified model and baseline methods on 11 VRPs provides promising and valuable baselines to the research community. Most of these VRPs have not been studied yet using neural solvers but are commonly used in real-world applications [1][2].
>
> Therefore, we believe our proposed method is novel and could make a good contribution to neural combinatorial optimization.
>
> [1] Heuristics for multi-attribute vehicle routing problems: A survey and synthesis. European Journal of Operational Research, 2013.
>
> [2] The vehicle routing problem: State of the art classification and review. Computers \& industrial engineering, 2016.
>
> **Our attribute decomposition Vs. all-attribute learning Vs. single-attribute learning**
>
> Our work is different from simply learning all attributes together (all-attribute learning). As discussed in the main paper, we have conducted an experiment on directly training the unified model on OVRPBLTW with all attributes (ST\_OVRPBLTW) to demonstrate the benefit of attribute composition. Line 11 in **Tabel 4, Appendix B** lists the results of ST\_OVRPBTW. According to the results, ST\_OVRPBLTW can achieve promising performance on the problem it trained on (OVRPBLTW) as well as two VRP variants with similar attributes (OVRPBTW and OVRPLTW). However, its performance becomes extremely poor for the rest VRPs (from simple CVRP to VRPBTW). This observation suggests that ST\_OVRPBLTW is over-fitted to CVRPBLTW with all attributes and cannot generalize well to other variants with a subset of attributes. This comparison also confirms the effectiveness and usefulness of our proposed method for learning different VRP variants with attribute composition.
>
> > **W2. The baseline POMO is classic, but not the SOTA. Quite a number of subsequent works surpass it, such as Efficient Active Search and 'Simulation-guided Beam Search for Neural Combinatorial Optimization'. In this sense, the comparison is not that convincing.**
>
> We address this concern from the following two aspects:
>
> **1) State-of-the-art heuristics:** We have added the result of LKH3 in Tabel 2. We also want to mention that the HGS baseline used in this work is a well-known and very powerful heuristic solver that can obtain (nearly) optimal solutions for different VRP variants [1]. It has been widely used as the baseline method in research papers [2,3,4], and is also the benchmark algorithm in different competitions [5,6].
>
> [1] A hybrid genetic algorithm with adaptive diversity management for a large class of vehicle routing problems with time-windows. Computers \& Operations Research 2013.
>
> [2] Towards Omni-generalizable Neural Methods for Vehicle Routing Problems. ICML 2023.
>
> [3] Neural Combinatorial Optimization with Heavy Decoder: Toward Large Scale Generalization. NeurIPS 2023.
>
> [4] Rl4co: an extensive reinforcement learning for combinatorial optimization benchmark. Submitted to ICLR 2024.
>
> [5] DIMACS VRPTW Challenge 2021.
>
> [6] EURO Meets NeurIPS Vehicle Routing Competition, NeurIPS 2022.
>
> **2) State-of-the-art NCO methods:** We have conducted an additional comparison study with SGBS [4], which is a state-of-the-art post-hoc search method for construction-based NCO. According to the results reported in the revised Table 2, the performance of our proposed method can be further improved with SGBS.
>
> We also want to mention that our method is implemented based on the model structure of POMO, which is the current standard baseline for construction-based NCO. As our framework is model-agnostic, one can easily promote cross-problem performance with more advanced base models in the future.
>
> [4] Simulation-guided beam search for neural combinatorial optimization. NeurIPS 2022.

---

> ### Author Response · Authors · 2023-11-20
> **Response to Review kE7F part 2/3**
>
> > **W3. Although the authors reviewed and criticized 'Efficient Training of Multi-task Combinatorial Neural Solver with Multi-armed Bandits', it is the most relevant baseline to this paper. It is good to compare with it in some adapted or tailored experimental settings.**
>
> The mentioned work provides an effective approach for doing multi-task learning for different combinatorial optimization problems (Travelling Salesman, Capacitated Vehicle Routing, Orienteering, and Knapsack). However, their work and ours actually investigate two different research directions, and they are hard to be directly compared. Our work adopts the attribute composition to learn a unified model from a small number of VRPs with basic attributes (Time Windows, Open Routes, Backhauls, and Duration Limits), and the model can be used to solve a much larger number of VRPs in a zero-shot manner. In contrast, the mentioned work requires one decoder for each problem, therefore prohibiting the zero-shot generalization on new VRPs. How to combine these two methods for a more efficient MTL NCO solver could be an interesting future work.
>
>
> > **W4. In table 2, where the proposed method is inferior to other neural baselines should be analyzed and explained.**
>
>
> Our multi-task learning model is slightly poorer than POMO, which is directly trained on each specific task. This result is reasonable and acceptable because 1) on the one hand, our model trained on five VRPs simultaneously. The training instances used on each VRP are only one-fifth of the compared single-task learning models. 2) on the other hand, the single-task learning model overfits a single VRP with a given distribution, which is also commonly observed in multi-task learning in other fields [1]. There is always a tradeoff between specific-task performance and general performance in a given budget. Our model significantly promotes cross-problem performance (from over 20\% to less than 5\%) and reduces the training cost while slightly sacrificing the performance on some specific tasks.
>
> [1] A survey on multi-task learning. IEEE Transactions on Knowledge and Data Engineering, 2021.
>
>
> **Correlation among Attributes:** We have conducted additional experiments to show the correlation and conflicting among attributes. Similar to [3], we separately train our model on each pair of VRPs to investigate the pairwise correlation among tasks. As a result, we have a total $C(5,2) = 10$ pairwise combinations with respect to the $5$ basic VRPs (i.e., CVRP, VRPTW, OVRP, VRPB, and VRPL). The detailed experimental settings and results can be found in Appendix F. We also provide an anonymous link (https://drive.google.com/file/d/1CtOwsgayGxFURA8luqIfDLMeCAFOJC23/view?usp=sharing) for fast checking of the resulting correlation matrix, which measures the performance gap between each task pair and our unified model. The positive values represent a positive correlation between two VRPs, while negative values denote a conflict.
>
> The results reveal that the CVRP, being a fundamental routing problem, has a positive correlation with all the other problems except for OVRP. The OVRP is found to exhibit the most severe conflicts among the attributes followed closely by the VRPTW. It aligns with our intuition that the open route conflicts with the other VRPs that force every route back to the depot. Similarly, time windows pose additional constraints on the choosing of the next node, which may significantly influence generalization performance, as we will discuss in \textbf{W2}.
>
> [3] Which tasks should be learned together in multi-task learning? ICML 2020.
>
> **Distribution of Different VRPs:** We have also conducted a comparison of distributions of different VRPs on the two-dimensional reduction space of the decoder hidden layer (1000 samples for each VRP), where the results can be found in Appendix F as well as an anonymous link for fast checking  (https://drive.google.com/file/d/1syLwzAy9tRspsvjrPkqAT4vQ-xrR67HH/view?usp=sharing). According to the result, there is a clearly distinct distribution of VRPTW compared with others as the time window attribute poses a strong constraint over the route. In addition to VRPTW, OVRP also shows a different pattern compared to others followed by VRPB, as we have mentioned that OVRP does not force routes back to the depot. In contrast, CVRP and VRPL follow a very close distribution and overlap with each other in the majority of areas, which is also reflected in the close final distances. This can be attributed to the fact that our route length limit is set to 3 on the normalized coordinates, which is easily satisfied due to the tighter capacity constraint. The distribution analysis of the hidden layer aligns with the findings from our correlation matrix discussions.

---

> ### Author Response · Authors · 2023-11-20
> **Response to Review kE7F part 3/3**
>
> > **W5. In table 3, the generic traditional methods like LKH-3 and OR-tools should be included.**
>
>
> Thank you for this suggestion, we have added the result of LKH3 in Tabel 2. However, according to the user guide of OR-Tools (https://developers.google.com/optimization/examples), it does not support the problems listed in Table 3. We also want to mention that the HGS baseline used in this work is a well-known and very powerful heuristic solver that can obtain (nearly) optimal solutions for different VRP variants [1]. It has been widely used as the baseline method in research papers [2,3,4], and is also the benchmark algorithm in different competitions [5,6]. We observed similar results when comparing HGS, LKH3, and OR-Tools on XML100 instances as is listed in the Table below.
>
>
>
> [1] A hybrid genetic algorithm with adaptive diversity management for a large class of vehicle routing problems with time-windows. Computers \& Operations Research 2013.
>
> [2] Towards Omni-generalizable Neural Methods for Vehicle Routing Problems. ICML 2023.
>
> [3] Neural Combinatorial Optimization with Heavy Decoder: Toward Large Scale Generalization. NeurIPS 2023.
>
> [4] Rl4co: an extensive reinforcement learning for combinatorial optimization benchmark. Submitted to ICLR 2024.
>
> [5] DIMACS VRPTW Challenge 2021.
>
> [6] EURO Meets NeurIPS Vehicle Routing Competition, NeurIPS 2022.
>
>
> |          | BK Dis. | HGS (5 s) Dis. | HGS (5 s) Gap | LKH3 (60 s) Dis. | LKH3 (60 s) Gap | OR-Tools (60 s) Dis. | OR-Tools (60 s) Gap | POMO (< 0.1s) Dis. | POMO (< 0.1s) Gap | Ours (< 0.1s) Dis. | Ours (< 0.1s) Gap |
> |----------|---------|----------------|---------------|-------------------|-----------------|----------------------|--------------------|-------------------|------------------|--------------------|------------------|
> | SET 1    | 13179   | 13182          | 0.02%         | 13555             | 2.29%           | 14780                | 16.01%             | 16988             | 29.47%           | 13946              | 6.23%            |
> | SET 2    | 10738   | 10740          | 0.01%         | 10969             | 1.53%           | 11923                | 12.28%             | 14851             | 41.01%           | 11318              | 5.60%            |
> | SET 3    | 18996   | 19028          | 0.17%         | 19993             | 2.76%           | 20791                | 12.31%             | 20767             | 11.83%           | 19867              | 4.44%            |
> | Average  | 14304   | 14317          | 0.07%         | 14839             | 2.19%           | 15831                | 13.53%             | 17535             | 27.44%           | 15044              | 5.42%            |

---

> > ### Comment · Reviewer_kE7F · 2023-11-23
> >
> > I appreciate the revisions made by the authors, which basically addressed my concerns. Therefore, I raised my score. I suggest the authors to clearly state their novelty and contribution in the methodology, e.g., multi-task learning, in the final version.

---

> > > ### Author Response · Authors · 2023-11-23
> > > **Thank you**
> > >
> > > Sincerely thank you for your response and support. We are very glad to know all your concerns have been addressed, and we will carefully revise our work to clearly state the novelty and contribution early in the paper.

---

> ### Author Response · Authors · 2023-11-22
>
> Thank you again for your time and effort in reviewing our work. There is only less than 1 day left to the rebuttal deadline, and we sincerely want to know whether our responses can successfully address all your concerns. We are glad to continually improve our work to address them.

---

### Official Review · Reviewer_QmWB · 2023-10-29

**Soundness:** 3 good
**Presentation:** 3 good
**Contribution:** 3 good
**Rating:** 6
**Confidence:** 2

**Summary:**

This work focuses on multi-task learning for vehicle routing problems. It proposes to build a unified model to solve these related problems in an end-to-end way, which is extended from the attention model with a unified encoder-decoder framework and attribute composition. Experiments prove the effectiveness of the proposed method.

**Strengths:**

1. The proposed multi-task solution for the routing problem is promising and could benefit downstream tasks.
2. The whole framework is well presented with good writing.
3. Experiments prove effectiveness on several problems.

**Weaknesses:**

I'm not an expert on vehicle routing. And I list my concerns here for reference.

This work adopts multi-task learning to solve various problems. Are there any related tasks or conflict tasks that are affected by this paradigm? I cannot find any discussion on it.

**Questions:**

Please refer to the weakness section.

---

> ### Author Response · Authors · 2023-11-20
> **Response to Review QmWB**
>
> Thank you very much for your time and suggestions. To further address your concern, we conducted three set of experiments to analyze the correlation of different attributes in our framework and also highlight the difference between our target combinatorial optimization problems and multi-task learning for general purposes.
>
> **Correlation among Attributes:** Similar to [3], we separately train our model on each pair of VRPs to investigate the pairwise correlation among tasks. As a result, we have a total $C(5,2) = 10$ pairwise combinations with respect to the $5$ basic VRPs (i.e., CVRP, VRPTW, OVRP, VRPB, and VRPL). The detailed experimental settings and results can be found in Appendix F. We also provide an anonymous link (https://drive.google.com/file/d/1CtOwsgayGxFURA8luqIfDLMeCAFOJC23/view?usp=sharing) for fast checking of the resulting correlation matrix, which measures the performance gap between each task pair and our unified model. The positive values represent a positive correlation between two VRPs, while negative values denote a conflict.
>
> The results reveal that the CVRP, being a fundamental routing problem, has a positive correlation with all the other problems except for OVRP. The OVRP is found to exhibit the most severe conflicts among the attributes followed closely by the VRPTW. It aligns with our intuition that the open route conflicts with the other VRPs that force every route back to the depot. Similarly, time windows pose additional constraints on the choosing of the next node, which may significantly influence generalization performance, as we will discuss in **W2**.
>
> [3] Which tasks should be learned together in multi-task learning? ICML 2020.
>
> **Distribution of Different VRPs:** We have also conducted a comparison of distributions of different VRPs on the two-dimensional reduction space of the decoder hidden layer (1000 samples for each VRP), where the results can be found in Appendix F as well as an anonymous link for fast checking  (https://drive.google.com/file/d/1syLwzAy9tRspsvjrPkqAT4vQ-xrR67HH/view?usp=sharing). According to the result, there is a clearly distinct distribution of VRPTW compared with others as the time window attribute poses a strong constraint over the route. In addition to VRPTW, OVRP also shows a different pattern compared to others followed by VRPB, as we have mentioned that OVRP does not force routes back to the depot. In contrast, CVRP and VRPL follow a very close distribution and overlap with each other in the majority of areas, which is also reflected in the close final distances. This can be attributed to the fact that our route length limit is set to 3 on the normalized coordinates, which is easily satisfied due to the tighter capacity constraint. The distribution analysis of the hidden layer aligns with the findings from our correlation matrix discussions.
>
>
> **Influence of the Degree of Constraint:** In the second part of the experiments, we examine how the degree of constraint affects the correlation and zero-shot generalization performance. We illustrate this using an example of VRPTW (Vehicle Routing Problem with Time Windows). Specifically, we trained a single-task model on VRPTW (ST\_VRPTW) with various TW lengths $\Delta_i$ = {0.15, 1 ,2 ,4} (where a smaller $\Delta_i$ represents a tighter TW constraint). We then evaluated the models on OVRPLTW and OVRPBTW. Our results reveal that as the TW length increases (i.e., the time window constraint is relaxed), the performance gap becomes larger compared to our multi-task learning approach. This can be attributed to the fact that the time window constraint, which is tightly enforced in our paper, is a dominant factor compared to other attributes that explain the strong generalization of ST\_VRPTW on tasks with time window constraints.
>
> |             | ST\_VRPTW  (0.15)   | ST\_VRPTW  (1.0)  | ST\_VRPTW (2.0)   | ST\_VRPTW (4.0)  |     MT    |
> |-------------|-----------|-----------|-----------|-----------|-----------|
> | OVRPLTW     |  7.83%    | 10.71%    | 14.85%    | 17.07%    |   7.71%   |
> | OVRPBTW     |  7.99%    | 11.55%    | 15.46%    | 17.46%    |   7.95%   |

---

> > ### Author Response · Authors · 2023-11-22
> >
> > Thank you again for your time and effort in reviewing our work. There is only less than 1 day left to the rebuttal deadline, and we sincerely want to know whether our responses can successfully address all your concerns. We are glad to continually improve our work to address them.

---

### Official Review · Reviewer_xWGs · 2023-10-30

**Soundness:** 3 good
**Presentation:** 4 excellent
**Contribution:** 3 good
**Rating:** 8
**Confidence:** 5

**Summary:**

This paper proposes a novel learning-based method to tackle cross-problem generalization in vehicle routing problems (VRPs), where the VRP variants are regarded as different combinations of a set of shared underlying attributes and solved by multi-task reinforcement learning. The experiments show the promising performance on unseen VRPs.

**Strengths:**

Cross-problem generalization is an important challenge for neural combinatorial optimization. This paper is an inspiring attempt at this leading direction.

**Weaknesses:**

A fly in the ointment is that the proposed method might be a little simple. Thus, I encourage the authors to explore more contributions on the methodology, so as to further improve the performance.

**Questions:**

1. This paper emphasizes zero-shot generalization, e.g., in the title. More discussions in experiments refer to fine-tuning, which seems to be inconsistent with the “zero-shot”. It is better to give more descriptions about this point.
2. As illustrated, different VRPs need to use different masking mechanisms to handle constraints. If a VRP variant is very complex, would it be hard to design a masking mechanism? Or even would there be some complex constraints that cannot be handled directly by masking?
3. Some typos still exist, such as the beginning of Section 3 and the reference “Select and optimize: Learning to aolve large-scale tsp instances”. Please check carefully.

---

> ### Author Response · Authors · 2023-11-20
> **Response to Review xWGs**
>
> Thank you very much for your time and effort in reviewing our work and providing valuable suggestions. We address your concerns as follows.
>
> > **Q1. This paper emphasizes zero-shot generalization, e.g., in the title. More discussions in experiments refer to fine-tuning, which seems to be inconsistent with the “zero-shot”. It is better to give more descriptions about this point.**
>
> Thank you for your valuable suggestion. We have reorganized and revised the content to emphasize our zero-shot generalization performance. Our evaluation includes the results on Table 2, Table 3, and most of Table 4, which are all assessed in a zero-shot manner. In addition, we have also included the results obtained through a powerful post-hoc search method (SGBS [1]) with the non-fine-tuned model for further comparison.
>
> [1] Simulation-guided beam search for neural combinatorial optimization. NeurIPS 2022.
>
> > **Q2. As illustrated, different VRPs need to use different masking mechanisms to handle constraints. If a VRP variant is very complex, would it be hard to design a masking mechanism? Or even would there be some complex constraints that cannot be handled directly by masking?**
>
> Yes, there are attributes that may not fit perfectly to the masking procedure. However, we argue that 1) our model can be extended to handle a large majority of problems through our attribute decomposition since the top ten attributes contribute to over 90\% of the VRP variants [1]. 2) If there are special attributes that can not be handled with the masking procedure, a specific decoding procedure might be required.
>
> [1] The vehicle routing problem: State of the art classification and review. Computers \& industrial engineering, 2016.
>
> > **Q3. Some typos still exist, such as the beginning of Section 3 and the reference “Select and optimize: Learning to aolve large-scale tsp instances”. Please check carefully.**
>
> Thank you very much for pointing them out. We have revised the main paper as well as the appendix to correct the typos carefully.

---

> > ### Author Response · Authors · 2023-11-22
> >
> > Thank you again for your time and effort in reviewing our work. There is only less than 1 day left to the rebuttal deadline, and we sincerely want to know whether our responses can successfully address all your concerns. We are glad to continually improve our work to address them.

---

> > > ### Comment · Reviewer_xWGs · 2023-11-23
> > >
> > > Thanks for your responses. I keep my rating.

---

> > > > ### Author Response · Authors · 2023-11-23
> > > > **Thank you**
> > > >
> > > > Sincerely thank you for your response and support. We will carefully revise and improve our work following your suggestions.

---

### Official Review · Reviewer_Cmb6 · 2023-11-01

**Soundness:** 1 poor
**Presentation:** 2 fair
**Contribution:** 2 fair
**Rating:** 3
**Confidence:** 4

**Summary:**

This paper proposes a unified model for solving routing problems with cross-constraints in a zero-shot manner. The model is based on multi-task learning and can effectively solve VRPs with different underlying attributes. The authors compare the performance of their model to single-task models trained specifically for each problem and show that their approach outperforms these models. The potential real-world applications of this approach to solving VRPs are also discussed.

**Strengths:**

1. The paper proposes a novel approach to solving VRPs using a unified model based on multi-task learning, which can effectively solve diverse VRPs in a zero-shot manner.
2. The authors provide experimental results on eleven VRPs to demonstrate the effectiveness of their approach. They compare the performance of their model to single-task models trained specifically for each problem and show that their approach outperforms these models.

**Weaknesses:**

1. The paper seems to simply add some VRP attributes as input to POMO and introduce an REINFORCE loss as multi-task loss. The novelty and contribution are relatively low. More analysis on the relations and effects of learning different tasks is expected.
2. In Table 2, the performance of LKH3 should be given for VRP variants except for CVRP.
3. The problem scale is relatively small, only up to 100.
4. In Table 3, for evaluation results on VRPBTW, POMO-VRPTW outperforms the proposed method. The result is not logical since the proposed method learns more features and information but leading to inferior performance in comparison with POMO-VRPTW. More analysis on this result is expected.
5. For all the VRP variants in Table 2 and Table 3, the baselines may not include all the corresponding state-of-the-art methods for multiple problems.

**Questions:**

1. How do different VRP attributes collaborate in the learning process?
2. Are the baseline methods in Table 1 and Table 2 are. the state-of-the-art methods for all VRP variants?
3. How does the proposed method perform on more large-scale problems?
4. Is the proposed method model-agnostic to other neural methods?
5. How does the proposed method encounters unseen attributes? Could the method be adapted quickly to new attributes with relatively small training samples?

---

> ### Author Response · Authors · 2023-11-20
> **Response to Review Cmb6 part 1/5**
>
> Thank you very much for your time and effort in reviewing our work and providing valuable suggestions. We address your concerns as follows.
>
> > **W1a The paper seems to simply add some VRP attributes as input to POMO and introduce an REINFORCE loss as multi-task loss. The novelty and contribution are relatively low.**
>
> In this work, rather than the simple multi-task learning, we investigate the **attribute composition for NCO**, which is a significant but less explored nature of many combinatorial optimization problems such as VRPs [1]. By only learning from a small number of problems with five fundamental attributes (capacity, time windows, open routes, backhauls, and duration limits), our model can easily generalize to tackle a larger number of problems with various combinations of these attributes in a zero-shot manner. This zero-shot generalization property is significantly different from a simple MTL problem and has not been investigated for neural combinatorial optimization. Our proposed attribute composition surpasses simply training all attributes as well as single-task learning in zero-shot generalization.
>
> In addition, the implementation of both our unified model and baseline methods on 11 VRPs provides promising and valuable baselines to the research community. Most of these VRPs have not been studied yet using neural solvers but are commonly used in real-world applications [1][2].
>
> Therefore, we believe our proposed method is novel and could make a good contribution to neural combinatorial optimization.
>
> [1] Heuristics for multi-attribute vehicle routing problems: A survey and synthesis. European Journal of Operational Research, 2013.
>
> [2] The vehicle routing problem: State of the art classification and review. Computers \& industrial engineering, 2016.

---

> ### Author Response · Authors · 2023-11-20
> **Response to Review Cmb6 part 2/5**
>
> > **W1b More analysis on the relations and effects of learning different tasks is expected.**
>
>
> Thank you for your valuable suggestion. We have conducted three sets of new experiments to further analyze the relation and effects of learning different tasks:
>
> **Correlation among Attributes:** Similar to [3], we separately train our model on each pair of VRPs to investigate the pairwise correlation among tasks. As a result, we have a total $C(5,2) = 10$ pairwise combinations with respect to the $5$ basic VRPs (i.e., CVRP, VRPTW, OVRP, VRPB, and VRPL). The detailed experimental settings and results can be found in Appendix F. We also provide an anonymous link (https://drive.google.com/file/d/1CtOwsgayGxFURA8luqIfDLMeCAFOJC23/view?usp=sharing) for fast checking of the resulting correlation matrix, which measures the performance gap between each task pair and our unified model. The positive values represent a positive correlation between two VRPs, while negative values denote a conflict.
>
> The results reveal that the CVRP, being a fundamental routing problem, has a positive correlation with all the other problems except for OVRP. The OVRP is found to exhibit the most severe conflicts among the attributes followed closely by the VRPTW. It aligns with our intuition that the open route conflicts with the other VRPs that force every route back to the depot. Similarly, time windows pose additional constraints on the choosing of the next node, which may significantly influence generalization performance, as we will discuss in **W2**.
>
> [3] Which tasks should be learned together in multi-task learning? ICML 2020.
>
> **Distribution of Different VRPs:** We have also conducted a comparison of distributions of different VRPs on the two-dimensional reduction space of the decoder hidden layer (1000 samples for each VRP), where the results can be found in Appendix F as well as an anonymous link for fast checking  (https://drive.google.com/file/d/1syLwzAy9tRspsvjrPkqAT4vQ-xrR67HH/view?usp=sharing). According to the result, there is a clearly distinct distribution of VRPTW compared with others as the time window attribute poses a strong constraint over the route. In addition to VRPTW, OVRP also shows a different pattern compared to others followed by VRPB, as we have mentioned that OVRP does not force routes back to the depot. In contrast, CVRP and VRPL follow a very close distribution and overlap with each other in the majority of areas, which is also reflected in the close final distances. This can be attributed to the fact that our route length limit is set to 3 on the normalized coordinates, which is easily satisfied due to the tighter capacity constraint. The distribution analysis of the hidden layer aligns with the findings from our correlation matrix discussions.
>
>
> **Influence of the Degree of Constraint:** In the second part of the experiments, we examine how the degree of constraint affects the correlation and zero-shot generalization performance. We illustrate this using an example of VRPTW (Vehicle Routing Problem with Time Windows). Specifically, we trained a single-task model on VRPTW (ST\_VRPTW) with various TW lengths $\Delta_i$ = {0.15, 1 ,2 ,4} (where a smaller $\Delta_i$ represents a tighter TW constraint). We then evaluated the models on OVRPLTW and OVRPBTW. Our results reveal that as the TW length increases (i.e., the time window constraint is relaxed), the performance gap becomes larger compared to our multi-task learning approach. This can be attributed to the fact that the time window constraint, which is tightly enforced in our paper, is a dominant factor compared to other attributes that explain the strong generalization of ST\_VRPTW on tasks with time window constraints.
>
> |             | ST\_VRPTW  (0.15)   | ST\_VRPTW  (1.0)  | ST\_VRPTW (2.0)   | ST\_VRPTW (4.0)  |     MT    |
> |-------------|-----------|-----------|-----------|-----------|-----------|
> | OVRPLTW     |  7.83%    | 10.71%    | 14.85%    | 17.07%    |   7.71%   |
> | OVRPBTW     |  7.99%    | 11.55%    | 15.46%    | 17.46%    |   7.95%   |

---

> ### Author Response · Authors · 2023-11-20
> **Response to Review Cmb6 part 3/5**
>
> > **W2. In Table 2, the performance of LKH3 should be given for VRP variants except for CVRP.**
>
> Thank you for your suggestion. We have incorporated the results in the revised Table 2.
>
> > **W3. The problem scale is relatively small, only up to 100.**
>
> We understand and acknowledge your concern regarding the significance of performance on instances at a large scale. In order to address this concern, we have conducted additional experiments on two groups of instances: 1) randomly generated instances ranging in size from 100 to 500, and 2) benchmark instances from CVRPLib ranging in size from 30 to 1000, where the best-known solutions (BKS) provided by CVRPLIB were utilized as the baseline for comparison. The detailed experimental settings and results can be found in Appendix E in the revised paper.
>
> We also report the results in the following two tables for a quick check. In the CVRPLib experiment, the 100 instances in Benchmark X are divided into four groups based on their problem size. According to the results, as the problem size increases, our unified model performs much better than POMO. For example, on the X problem with size 700-100, our model achieves a $13.8\%$ optimal gap, while the gap for POMO is $31.9\%$.
>
> |             | N200 |         | N300 |         | N400 |         | N500 |         |
> |-------------|------|---------|------|---------|------|---------|------|---------|
> | HGS         | 21.82| /       | 25.76| /       | 28.14| /       | 32.01| /       |
> | LKH3        | 22.34| 2.39\%  | 26.45| 2.72\%  | 29.05| 3.23\%  | 32.07| 0.17\%  |
> | OR-tools    | 24.14| 10.63\% | 28.82| 11.89\% | 31.77| 12.89\% | 35.82| 11.90\% |
> |-------------|------|---------|------|---------|------|---------|------|---------|
> | POMO        | 23.11| 5.90\%  | 29.21| 13.42\% | 34.75| 23.48\% | 43.12| 34.68\% |
> |             | 22.85| 4.71\%  | 28.79| 11.77\% | 33.76| 19.98\% | 41.14| 28.50\% |
> | Ours        | 22.96| 5.25\%  | 28.42| 10.35\% | 32.53| 15.60\% | 38.53| 20.34\% |
> |             | 22.73| 4.18\%  | 28.07| 8.97\%  | 32.02| 13.78\% | 37.45| 16.99\% |
> | Ours SGBS   | **22.34**| **2.40\%** | **27.30**| **6.00\%** | **30.92**| **9.87\%** | **35.78**| **11.75\%** |
>
>
> | Benchmark | Size     | BKS      | POMO    | Ours   |
> |-----------|----------|----------|---------|--------|
> | A         | 31-79    | 1041.9   | 1104.8  | 1066.8 |
> | B         | 30-77    | 963.7    | 1065.7  | 992.2  |
> | F         | 44-134   | 707.7    | 770.6   | 760.6  |
> | M         | 100-199  | 1083.6   | 1145.2  | 1141.9 |
> | P         | 15-100   | 587.4    | 660.4   | 627.1  |
> | X         | 100-300  | 33868.5  | 36299.07| 35954.7|
> | X         | 300-500  | 63774.8  | 71524.15| 68841.6|
> | X         | 500-700  | 88561.4  | 107927.9| 98843.8|
> | X         | 700-1000 | 113619.5 | 149858.6| 129270.9|
>
> In addition, we want to acknowledge that the VRP variant with 100 nodes can have practical value in real-world applications as well as academic research. For example, all the 10,000 instances used for testing machine learning heuristics in the AAAI 2022 ML4OR workshop are of size 100 [1]. In addition, there are several recent works specific to large-scale generalization for construction-based NCO [2,3]. Our multi-task learning approach can be readily incorporated into that model for enhanced performance on large-scale problems.
>
> [1] 10,000 optimal CVRP solutions for testing machine learning based heuristics. ML4OR workshop AAAI 2022.
>
> [2] BQ-NCO: Bisimulation Quotienting for Efficient Neural Combinatorial Optimization. NeurIPS 2023 (to appear).
>
> [3] Neural Combinatorial Optimization with Heavy Decoder: Toward Large Scale Generalization. NeurIPS 2023 (to appear).
>
> > **W4. In Table 3, for evaluation results on VRPBTW, POMO-VRPTW outperforms the proposed method. The result is not logical since the proposed method learns more features and information but leading to inferior performance in comparison with POMO-VRPTW. More analysis on this result is expected.**
>
>
> As we have mentioned in **W1b**, we believe the reason is that the time window constraint used in our paper is much tighter than other attributes. As a result, ST\_VRPTW generalizes well on tasks with time window constraints. We conducted experiments on training single-task model on VRPTW ST\_VRPTW with different TW lengths $\Delta_i$ = \{0.15, 1, 2 ,4\} (small $\Delta_i$ represents tight TW constraint) and test the models on OVRPLTW and OVRPBTW. The results show that the gap becomes much worse compared to our multi-task learning with the increasing TW length (i.e., relaxing time window constraint).
>
> |             | ST\_VRPTW  (0.15)   | ST\_VRPTW  (1.0)  | ST\_VRPTW (2.0)   | ST\_VRPTW (4.0)  |     MT    |
> |-------------|-----------|-----------|-----------|-----------|-----------|
> | OVRPLTW     |  7.83%    | 10.71%    | 14.85%    | 17.07%    |   7.71%   |
> | OVRPBTW     |  7.99%    | 11.55%    | 15.46%    | 17.46%    |   7.95%   |

---

> ### Author Response · Authors · 2023-11-20
> **Response to Review Cmb6 part 4/5**
>
> > **W5. For all the VRP variants in Table 2 and Table 3, the baselines may not include all the corresponding state-of-the-art methods for multiple problems.**
>
> We address this concern from the following two aspects:
>
> **1) State-of-the-art heuristics:** We have added the result of LKH3 in Table 2. We also want to mention that the HGS baseline used in this work is a well-known and very powerful heuristic solver that can obtain (nearly) optimal solutions for different VRP variants. It has been widely used as the baseline method in research papers [1,2,3], and is also the benchmark algorithm in different competitions [4,5].
>
> [1] Towards Omni-generalizable Neural Methods for Vehicle Routing Problems. ICML 2023.
>
> [2] Neural Combinatorial Optimization with Heavy Decoder: Toward Large Scale Generalization. NeurIPS 2023.
>
> [3] Rl4co: an extensive reinforcement learning for combinatorial optimization benchmark. Submitted to ICLR 2024.
>
> [4] DIMACS VRPTW Challenge 2021.
>
> [5] EURO Meets NeurIPS Vehicle Routing Competition, NeurIPS 2022.
>
> **2) State-of-the-art NCO methods:** We have conducted an additional comparison study with the simulation-guided bean search (SGBS) [4], which is a state-of-the-art post-hoc search method for construction-based NCO. According to the results reported in the revised Table 2, the performance of our proposed method can be further improved with SGBS.
>
> We also want to mention that our method is implemented based on the model structure of POMO, which is the current standard baseline for construction-based NCO. As our framework is model-agnostic, one can easily promote cross-problem performance with more advanced base models in the future.
>
> [4] Simulation-guided beam search for NCO. NeurIPS 2022.
>
> | Problem | Method | N=50 Dis. | N=50 Gap | N=50 Time | N=100 Dis. | N=100 Gap | N=100 Time |
> | --- | --- | --- | --- | --- | --- | --- | --- |
> | CVRP | HGS | 10.38 | - | 7h | 15.54 | - | 14h |
> |  | LKH3 | 10.38 | 0.00% | 7h | 15.61 | 0.46% | 14h |
> |  | AM (Samp1280) | 10.59 | 2.02% | 7m | 16.16 | 4.00% | 30m |
> |  | MDAM (BS50) | 10.48 | 0.96% | 7.5m | 15.99 | 2.90% | 26m |
> |  | GCAM (Samp1280) | 10.64 | 2.50% | - | 16.29 | 4.83% | - |
> |  | POMO | 10.53 | 1.41% | 3s | 15.87 | 2.13% | 10s |
> |  | POMO (Aug8) | 10.44 | 0.58% | 15s | 15.75 | 1.36% | 1.1m |
> |  | SGBS | 10.39 | 0.12% | 2.0m | 15.63 | 0.62% | 11.8m |
> |  | Ours | 10.56 | 1.73% | 3s | 15.90 | 2.29% | 11s |
> |  | Ours (Aug8) | 10.47 | 0.85% | 20s | 15.80 | 1.71% | 1.2m |
> |  | Ours+SGBS | 10.40 | 0.18% | 2.3m | 15.66 | 0.81% | 12.6m |
> | VRPTW | HGS | 16.30 | - | 7h | 26.14 | - | 14h |
> |  | LKH3 | 16.52 | 1.36% | 7h | 26.60 | 1.76% | 14h |
> |  | DRL (BS10) | 17.90 | 9.82% | 1m | 29.50 | 12.85% | 2m |
> |  | DRL (BS10) +LNS | 16.94 | 3.93% | 11m | 27.44 | 4.97% | 65m |
> |  | POMO | 16.78 | 2.97% | 3s | 27.13 | 3.77% | 11s |
> |  | POMO (Aug8) | 16.66 | 2.22% | 19s | 26.91 | 2.93% | 1.2m |
> |  | SGBS | 16.55 | 1.52% | 2.9m | 26.55 | 1.58% | 15.1m |
> |  | Ours | 16.96 | 4.06% | 3s | 27.46 | 5.05% | 11s |
> |  | Ours (Aug8) | 16.80 | 3.09% | 20s | 27.13 | 3.81% | 1.2m |
> |  | Ours+SGBS | 16.58 | 1.71% | 3.2m | 26.63 | 1.89% | 17.9m |
> | OVRP | HGS | 6.49 | - | 7h | 9.71 | - | 14h |
> |  | LKH3 | 6.52 | 0.46% | 7h | 9.75 | 0.41% | 14h |
> |  | POMO | 6.73 | 3.67% | 3s | 10.18 | 4.91% | 10s |
> |  | POMO (Aug8) | 6.63 | 2.14% | 16s | 10.07 | 3.76% | 1.1m |
> |  | SGBS | 6.56 | 1.12% | 2.1 m | 9.89 | 1.92% | 12.1m |
> |  | Ours | 6.81 | 4.90% | 3s | 10.34 | 6.56% | 11s |
> |  | Ours (Aug8) | 6.71 | 3.40% | 20s | 10.14 | 4.48% | 1.2m |
> |  | Ours+SGBS | 6.59 | 1.58% | 2.5m | 9.94 | 2.38% | 13.4m |
> | VRPB | HGS | 7.69 | - | 7h | 11.13 | - | 14h |
> |  | LKH3 | 7.70 | 0.18% | 7h | 11.29 | 1.40% | 14h |
> |  | POMO | 7.92 | 3.06% | 3s | 11.57 | 3.88% | 10s |
> |  | POMO (Aug8) | 7.84 | 2.05% | 15s | 11.43 | 2.68% | 1.1m |
> |  | SGBS | 7.78 | 1.22% | 1.9m | 11.31 | 1.59% | 11m |
> |  | Ours | 8.17 | 6.36% | 3s | 11.72 | 5.23% | 11s |
> |  | Ours (Aug8) | 7.87 | 2.40% | 20s | 11.53 | 3.58% | 1.2m |
> |  | Ours+SGBS | 7.78 | 1.25% | 2.1m | 11.36 | 2.06% | 11.7m |
> |  VRPL  |   HGS   | 10.37 |  -   |  7h  | 15.54 |   -   |  14h  |
> |        |  LKH3   | 10.37 |0.03% |  7h  | 15.61 |0.43%  |  14h  |
> |        |  POMO   | 10.55 |1.78% |  3s  | 15.84 |1.96%  |  10s  |
> |        |POMO(Aug8)| 10.46 |0.91% | 16s  | 15.72 |1.14%  |  1.1m |
> |        |  SGBS   | 10.40 |0.30% | 2.3m | 15.64 |0.66%  | 13.1m |
> |        |  Ours   | 10.56 |1.88% |  3s  | 15.96 |2.72%  |  11s  |
> |        |Ours(Aug8)| 10.47 |0.98% | 20s  | 15.80 |1.66%  |  1.2m |
> |        |Ours+SGBS | 10.40 |0.33% | 2.6m | 15.67 |0.83%  | 14.3m |
> | Average|  POMO   | 10.50 |2.58% |  3s  | 16.12 |3.33%  |  10s  |
> |        |POMO(Aug8)| 10.41 |1.58% | 16s  | 15.97 |2.37%  |  1.1m |
> |        |  SGBS   | 10.34 |0.86% |2.24m | 15.81 |1.28%  | 12.6m |
> |        |  Ours   | 10.61 |3.78% |  3s  | 16.27 |4.37%  |  11s  |
> |        |Ours(Aug8)| 10.46 |2.14% | 20s  | 15.81 |3.05%  |  1.2m |
> |        |Ours+SGBS | 10.35 |1.01% | 2.5m | 15.85 |1.59%  | 14.0m |

---

> ### Author Response · Authors · 2023-11-20
> **Response to Review Cmb6 part 5/5**
>
> > **Q1. How do different VRP attributes collaborate in the learning process?**
>
> Please refer to our response to \textbf{W1a} and **W1b**.
>
> > **Q2. Are the baseline methods in Table 1 and Table 2 are. the state-of-the-art methods for all VRP variants?**
>
> Please refer to our response to **W5**.
>
> > **Q3. How does the proposed method perform on more large-scale problems?**
>
>
> Please refer to our response to **W3**.
>
>
> > **Q4.Is the proposed method model-agnostic to other neural methods?**
>
>
> Yes, our method is model agnostic to both 1) the model structures and 2) the post-hoc neural search methods. For the former, we can easily incorporate the attribute composition input and multi-task training procedure into the new model structure. For the latter, we can directly adopt the post-hoc search approach with our method, such as the powerful SGBS approach in response to **W5**.
>
>
> > **Q5. How does the proposed method encounters unseen attributes? Could the method be adapted quickly to new attributes with relatively small training samples?**
>
> If the new attribute is closely relative to the seen one (e.g., flexible soft time windows), it can be properly handled by the unified model with a fast adaption. However, for a brand new and irrelevant attribute, we should expect our proposed method will require more effort to handle it, which might involve attribute-dependent inference design and model retraining.
>
> In addition, our model can be extended to handle a large majority of problems through our attribute decomposition since the top ten attributes contribute to over 90\% of the VRP variants [1].
>
> [1] The vehicle routing problem: State of the art classification and review. Computers \& industrial engineering, 2016.

---

> ### Author Response · Authors · 2023-11-22
>
> Thank you again for your time and effort in reviewing our work. There is only less than 1 day left to the rebuttal deadline, and we sincerely want to know whether our responses can successfully address all your concerns. We are glad to continually improve our work to address them.

---

> > ### Author Response · Authors · 2023-11-23
> >
> > Thank you again for your time and effort in reviewing our work. There are only less than 2 hours left to the rebuttal deadline, and we sincerely want to know whether our responses can successfully address all your concerns. We are more than happy to provide quick responses to any unaddressed concerns.

---

### Author Response · Authors · 2023-11-20
**General Response**

We sincerely thank all reviewers for their constructive comments and valuable suggestions. We are also glad to know the reviewers find our proposed method well-motivated (kE7F), inspiring (xWGs), promising (QmWB), and can effectively solve 11 different VRP variants (Cmb6, xWGs, QmWB, kE7F) where the cross-problem generalization is an important challenge for neural combinatorial optimization (xWGs).

Following their suggestions, we have carefully revised our paper and the major revision can be summarized as follows:

+ **Correlation among Attributes:** We have conducted new experiments on 10 pairwise multi-task learning using our unified model to analyze the correlation and interaction of attributes. We also extract the hidden layer from the decoder of our unified model and compare different VRPs in the two-dimensional manifold. The results, analyses, and discussion on generalization performance can be found in **Appendix F**.


+ **Generalization to Large-Scale Problem:** We have also tested our proposed model on large-scale randomly generated instances and CVRPLib test suites with sizes ranging from 30 to 1,000. The results and analyses can be found in **Appendix E**.

+ **More Comprehensive Comparison:** We have integrated the simulation-guided beam search for comparison and promoting zero-shot generalization performance of our unified model. The results of HGS and LKH3 are included as well. The results have been added to the revised Table 2, and the discussion can be found in **Subsection 5.1**.


All major changes are highlighted in blue in the revised paper, and point-to-point responses can be found below for each reviewer. We are also glad to continually improve our work to address any further concerns.

Best Regards,

Paper7171 Authors

---

### Meta-Review · Area_Chair_LmR5 · 2023-12-12

**Metareview:**

This paper proposes to use multi-task learning and composable attributes for zero-shot generalization in the context of vehicle routing problems. The main concerns raised by the reviewers were insufficient novelty and unclear gains over SOTA. The rebuttal included additional experiments, for example to analyze the correlation and interaction of attributes and more baseline comparisons. The final scores were mixed: one accept, two borderline accepts, and one reviewer arguing for rejection, stating that their concerns about novelty and performance remained after the author response. While zero-shot generalization is a very relevant problem for neural combinatorial optimization, the AC agrees with reviewer Cmb6 that the solution presented by the authors lacks methodological novelty, as attribute composition has been employed for zero-shot generalization (see, e.g., [Lampert et al, CVPR 2009]), and therefore the novelty of the paper lies on the application of existing concepts to NCO. Taking into account the performance of the approach and the novelty aspect, the AC considers the paper does not pass the acceptance bar of ICLR.

**Justification For Why Not Higher Score:**

The concerns raised by reviewer Cmb6 about novelty and performance of the proposed approach are legitimate.

**Justification For Why Not Lower Score:**

N/A

---

### Decision · Program_Chairs · 2024-01-16

Reject